



# Modeling snow slab avalanches caused by weak layer failure – Part I: Slabs on compliant and collapsible weak layers

Philipp L. Rosendahl[1,2,*] and Philipp Weißgraeber[1,3,4,*]

[1]2φ GbR, www.2phi.de, Tübingen, Germany
[2]Technische Universität Darmstadt, Department of Mechanical Engineering, Germany
[3]Robert Bosch GmbH, Corporate Research and Advance Engineering, Renningen, Germany
[4]ARENA2036 research campus, Universität Stuttgart, Germany
[*]Contributed equally to this work.

**Correspondence:** mail@2phi.de

**Abstract.** Dry-snow slab avalanche release is preceded by a fracture process within the snowpack. Recognizing weak layer collapse as an integral part of the fracture process is crucial and explains phenomena such as whumpf sounds and remote triggering of avalanches from low-angle terrain. In this two-part work we propose a novel closed-form analytical model for a snowpack under skier loading and a mixed-mode failure criterion for nucleation of weak layer failure.

In the first part of this two-part series we introduce a closed-form analytical model of a snowpack accounting for the deformable layer. Despite the importance of persistent weak layers for slab avalanche release, no simple analytical model accounting for weak layer deformations is available. The proposed model provides deformations of the snow slab, weak layer stresses and energy release rates of cracks within the weak layer. It generally applies to skier-loaded slopes as well as stability tests such as the propagation saw test. A validation with a numerical reference model shows very good agreement of the stress

and energy release rates results in several parametric studies including analyses of the bridging effect and slope angle dependence. The proposed model is used to analyze 93 propagation saw tests. Computed weak layer fracture toughness values are physically meaningful and in excellent agreement with finite element analyses.

    In the second part of the series we make use of the present mechanical model to establish a novel failure criterion crack nucleation in weak layers. The code used for the analyses in both parts is publicly available.

## 1 Introduction

Dry-snow slab avalanches can release when a persistent weak layer of, e.g., surface hoar or depth hoar breaks (see the well-known image of a partially collapsed weak layer by Jamieson and Schweizer (2000) shown in Figure 1). Weak layer failure can be triggered by additional loads like a skier. If the conditions allow for crack propagation, a triggered initial defect may extend across slopes and eventually cause the slab to fail and slide.

The earliest approaches to snowpack stability were so-called stability indices. They consider snowpack loading owing to the weight of the snow slab and owing to additional loading by a skier (Perla, 1977; Föhn, 1987). To account for snow stratification improved stability index models were proposed by Habermann et al. (2008) or Monti et al. (2015). However, these local models





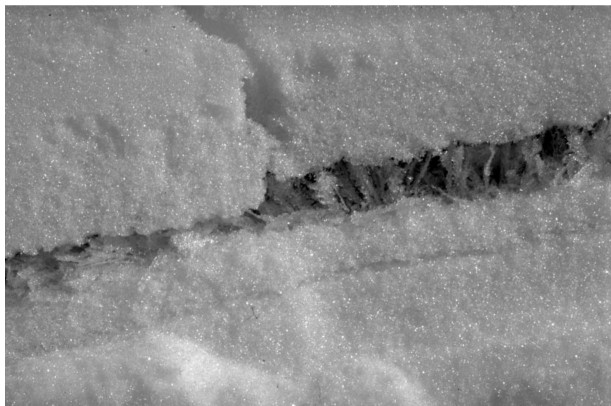

**Figure 1.** Weak layer of buried surface hoar. Left of the vertical slab fracture the weak layer has collapsed, whereas on the right hand side the porous weak layer is still intact. Reprinted from the Journal of Glaciology (Jamieson and Schweizer, 2000) with permission of the International Glaciological Society.

are insufficient to describe the stability of snowpacks across slopes (Bellaire and Schweizer, 2011). Many researchers suggested that these indices are incomplete as they do not account for the propagation of the failure within the snowpack (van Herwijnen and Jamieson, 2007).

To evaluate stability indices or fracture mechanics criteria, a model of the stress distribution within the snowpack and

especially the weak layer is needed. Using the exact solution for a concentrated load on a homogeneous semi-infinite plate, Föhn (1987) proposed a model for the stress distribution below a skier. A first interface model with shear for snowpacks with weak layers was proposed by McClung (1979). Linear elastic finite element analyses have shown that the effect of layering can play an important role (Schweizer, 1993; Habermann et al., 2008). This was accounted for in the model of Monti et al. (2015) by using an equivalent homogenized snow layer with effective uniform Young's modulus and a modified total slab thickness.

Shear-lag models consider only the extensional stiffness of the slab above the weak layer. To account for bending deformation, beam models using Euler-Bernoulli beam kinematics (Heierli and Zaiser, 2006) and Timoshenko beam kinematics (Heierli and Zaiser, 2008; Heierli et al., 2008) were proposed. The latter accounts for shear deformation of the slab and thus for the low shear stiffness of cohesive snow. In such models, the slab bending stiffness controls the load distribution. It increases in a cubic relation with slab thickness whereas the extensional stiffness considered in shear-lag models only changes proportionally

with slab thickness. Hence, beam models cover bridging effects (Schweizer and Camponovo, 2001; Schweizer and Jamieson, 2003; Thumlert and Jamieson, 2014). Bridging effects show that stiff and thick layers above a weak layer can distribute loads more evenly leading to smaller stresses in the weak layer. However, the effect of weak layer compliance in normal direction is unaccounted for in the aforementioned beam models (Heierli and Zaiser, 2008). The snowpack is assumed to rest on a rigid weak layer and only slab deformation in the region of a collapsed weak layer is modeled. Yet, it is known that the deformation

of the weak layer is crucial for deformations and the local load transfer in the snowpack (Reiweger and Schweizer, 2010). (Chiaia et al., 2008; Gaume et al., 2013) consider weak layer deformability only in shear.





In the first part of this two-part contribution, a novel modeling approach for the description of weak layer failure is given. It aims at providing a model that fully accounts for the weak layer's effect on deformations and load transfer and solutions to the mixed-mode energy release rates of cracks within the weak layer. The model is validated using finite element analyses and field experiments. In the second part we propose a new failure criterion, which physically links stability indices and fracture

mechanics models. Here, the necessary distinction of stress-strength and fracture mechanics approaches is highlighted and discussed.

## 2 Modeling approach

Deformation, stresses and consequently the energy release rate of cracks within the weak layer are controlled by loading and the complete stratigraphy of the snowpack. Deformations of the slab and in particular of the weak layer must be rendered

sufficiently accurate. The slab is loaded in local bending and stretching which we account for using beam and rod kinematics. As in the analysis by Heierli et al. (2008) we model the slab as an elastic beam with bending and shear deformation. In order to account for weak layer deformations, the present model rests the beam on an elastic foundation of an infinite set of springs with compressive and shear stiffness commonly referred to as Winkler foundation. Base layers are assumed rigid. The model provides slab deformations, weak layer normal and shear stresses as well as the energy release rate of cracks. Skier penetration

is not considered. Like other models given in literature and discussed above we employ linear elasticity and neglect, e.g., viscous or plastic deformations. Consequential limitations are discussed in Section 4.

### 2.1 Governing equations

Consider the snowpack model on an inclined slope of angle $\varphi$ depicted in Figure 2a. The plane strain beam representing the snow slab has an out-of-plane thickness $b$, height $h$ and length $l$. The weak layer thickness is denoted $t$. The slab is assumed

to be homogeneous with Young's modulus $E$, Poisson's ratio $\nu$ and density $\rho$. The snow slab is loaded by its own weight prescribed as distributed loads $q_t = \rho g h b \sin(\varphi)$ in tangential $x$-direction and $q_n = \rho g h b \cos(\varphi)$ in normal $z$-direction where $g$ is the acceleration of gravity. Additional loading (e.g., by the weight force of mass $m$ of a skier, snowmobile or others) is represented by a concentrated force $F = mgb/l_o$ where $l_o$ is the effective out-of-plane length of the object such as the length of skis. $F$ is split into normal and tangential concentrated forces $F_n = F \cos(\varphi)$ and $F_t = F \sin(\varphi)$, respectively. The weak layer

of thickness $t$ consists of an infinite set of springs with compressive stiffness $k_n = E_{\text{weak}} b/t$ and shear stiffness $k_t = G_{\text{weak}} b/t$ with the shear modulus $G_{\text{weak}} = E_{\text{weak}}/(2(1+\nu))$. Cracks within the weak layer are modeled by removing the support of the beam on the crack length $a$.

Timoshenko kinematics for the beam allow for shear deformation. Initially plane beam cross sections may rotate by an angle $\psi$ (see Figure 2b) yet remain plane during deformation, which yields the general deformation kinematics of the beam

$$u_z(x,z) = u(x) + z\psi(x), \quad w_z(x,z) = w(x), \tag{1}$$



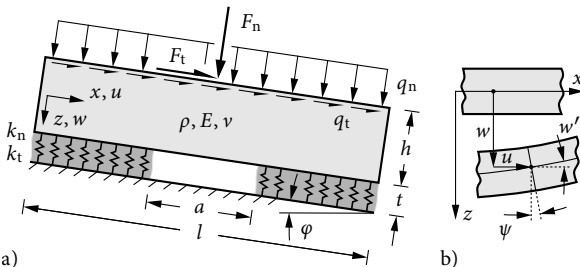

**Figure 2.** Snowpack modeled a) as a beam on elastic foundation of infinite set of shear and compressive springs b) using Timoshenko beam kinematics.

where $u$ is the horizontal displacement and $w$ the vertical deflection of the beam middle plane, respectively. The index $z$ is introduced to distinguish between midplane deformations $u(x)$ and $w(x)$ and the actual displacement fields $u_z(x,z)$ and $w_z(x,z)$.

Enforcing equilibrium of forces and moments and using the laws of elasticity allows for deriving a set of ordinary differential equations (ODEs) with constant coefficients which describe the deformation of the snow slab (see Appendix A). The horizontal displacement $u$ is obtained from

$$EAu''(x) - k_t u(x) + q_t = 0, \tag{2}$$

where $A = hb$ is the snow slab cross section. The vertical beam deflection $w$ and the rotation of the beam cross section $\psi$ are described by

$$w''''(x) - \frac{k_n}{\kappa GA}w''(x) + \frac{k_n}{EI}w(x) = \frac{q_n}{EI}, \tag{3}$$

$$\psi(x) = -\frac{EI}{\kappa GA}w'''(x) + \left(\frac{EI\,k_n}{(\kappa GA)^2} - 1\right)w'(x), \tag{4}$$

where $\kappa = 5/6$ is the shear correction factor for rectangular cross sections and $I = bh^3/12$ the moment of inertia with respect to the $y$-axis.

The general solution of ODE (2) for the horizontal displacement is given by

$$u(x) = c_1 \cosh(\mu x) + c_2 \sinh(\mu x) + \frac{q_t}{k_t}, \tag{5}$$

with the eigenvalue

$$\mu = \sqrt{\frac{k_t}{EA}}. \tag{6}$$

The free constants $c_1$ and $c_2$ must be determined from boundary conditions. The solution of the coupled ODEs (3) and (4) is of exponential type as well. Depending on the material parameters, the eigenvalues of this solution may become real or complex.





When $k_{\mathrm{n}}EI \geq \left(4(\kappa GA)^2\right)$ the eigenvalues are real numbers

$$\lambda_{1,2} = \frac{1}{\sqrt{2}} \sqrt{\frac{k_{\mathrm{n}}}{\kappa GA} \pm \sqrt{\left(\frac{k_{\mathrm{n}}}{\kappa GA}\right)^2 - \frac{4k_{\mathrm{n}}}{EI}}}, \tag{7}$$

and the general solution is given by

$$\begin{aligned}
w(x) &= c_3 \cosh(\lambda_1 x) + c_4 \sinh(\lambda_1 x) \\
&\quad + c_5 \cosh(\lambda_2 x) + c_6 \sinh(\lambda_2 x) + \frac{q_{\mathrm{n}}}{k_{\mathrm{n}}}.
\end{aligned} \tag{8}$$

If $k_{\mathrm{n}}EI < \left(4(\kappa GA)^2\right)$ the eigenvalues are complex and read

$$\lambda_{1,2}^* = \sqrt{\sqrt{\frac{k_{\mathrm{n}}}{4EI}} \pm \frac{k_{\mathrm{n}}}{4\kappa GA}}. \tag{9}$$

In this case the general solution of the normal deflection is given by

$$\begin{aligned}
w(x) &= \mathrm{e}^{\lambda_1^* x} \left(c_3 \cos(\lambda_2^* x) + c_4 \sin(\lambda_2^* x)\right) \\
&\quad + \mathrm{e}^{-\lambda_1^* x} \left(c_5 \cos(\lambda_2^* x) + c_6 \sin(\lambda_2^* x)\right) + \frac{q_{\mathrm{n}}}{k_{\mathrm{n}}},
\end{aligned} \tag{10}$$

Again, the constants $c_3$ to $c_6$ must be determined from boundary conditions.

For regions without elastic foundation, e.g., above a crack or when modeling propagation saw tests (PST), we obtain $k_{\mathrm{t}} = k_{\mathrm{n}} = 0$ and the differential equations simplify to

$$EA u''(x) + q_{\mathrm{t}} = 0, \tag{11}$$

$$EI w''''(x) = q_{\mathrm{n}}, \tag{12}$$

$$\psi(x) = -\frac{EI}{\kappa GA} w'''(x) - w'(x). \tag{13}$$

The corresponding solutions are given by

$$u(x) = c_7 + c_8 x - \frac{q_{\mathrm{t}}}{2EA} x^2, \tag{14}$$

$$w(x) = c_9 + c_{10} x + c_{11} x^2 + c_{12} x^3 + \frac{q_{\mathrm{n}}}{24EI} x^4. \tag{15}$$

As before, the free constants must be determined from boundary conditions which are defined by the final assembly of the solution.

## 2.2 Assembling the solution

The present model readily applies to an inclined skier-loaded snowpack with or without crack in the weak layer as well as to propagation saw tests. Therefore, considered snowpack configurations must be assembled from general solutions of beam





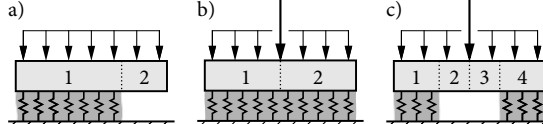

**Figure 3.** Snowpack configurations assembled from beam segments with boundary and transmission conditions: a) PST b) skier load on intact weak layer c) skier load on weak layer with crack. Weak layer cracks are modeled by removing support of the beam. For the sake of clarity only vertical loads are shown.

segments with or without elastic foundation given above. The free constants are determined from boundary and transmission conditions.

To model a propagation saw test two beam segments are required as shown in Figure 3a. The left part of the snow slab rests on an elastic foundation representing the intact weak layer. The right part is a cantilever beam where cutting has removed the
weak layer support. The slab is loaded by its own weight $q_\mathrm{n}$ and $q_\mathrm{t}$. Free left and right ends require vanishing section forces and moments. Using the expressions

$$M = EI\psi', \quad Q = \kappa GA\left(w' + \psi\right), \quad N = EAu', \tag{16}$$

for the bending moment $M$, the lateral shear force $Q$ and the normal force $N$ yields

$$u' = 0, \quad \psi' = 0, \quad w' + \psi = 0, \tag{17}$$

at free ends. At the transition between the unsupported and supported segments of displacements, cross section rotation, section forces and moments must be $C^0$-continuous. This yields

$$u_\mathrm{l} = u_\mathrm{r}, \quad w_\mathrm{l} = w_\mathrm{r}, \quad \varphi_\mathrm{l} = \varphi_\mathrm{r},$$
$$u_\mathrm{l}' = u_\mathrm{r}', \quad w_\mathrm{l}' = w_\mathrm{r}', \quad \varphi_\mathrm{l}' = \varphi_\mathrm{r}', \tag{18}$$

where the indices 'l' and 'r' denote quantities left and right of the discontinuity, respectively.

In skier loaded snowpacks (Figs. 3b and 3c) the skier point load adds discontinuities of normal and transverse shear forces and the solution is assembled from two or four regions. Skier loaded slopes extend far beyond the influence zone of local skier loading. Hence, we have chosen a length of 25 times the thickness of the slab ($l = 25h$) to avoid any edge effects. Of course, also shorter slabs can be modeled to account for the resulting edge effects on the mechanical response if necessary. Again, at either end of the assembly boundary conditions of vanishing section forces hold. At crack tips the continuity conditions
Eq. (18) hold. At the concentrated skier load the transmission conditions must be modified according to

$$u_\mathrm{l}' = u_\mathrm{r}' + \frac{F_\mathrm{t}}{EA}, \qquad w_\mathrm{l}' = w_\mathrm{r}' + \frac{F_\mathrm{n}}{\kappa GA}. \tag{19}$$

The boundary and transmission conditions for the respective load case provide a linear system of equations with up to 24 unknown constants. The system can be solved easily using any mathematical toolbox. Closed-form solutions for $u$, $w$ and $\psi$ can then be given in piecewise form.





## 2.3 Weak layer stresses

Since the weak layer is represented as an elastic Winkler foundation, also known as weak interface model (Lenci, 2001), it is not modeled as a complete continuum. Instead, it only transfers compressive and lateral shear loads. In this simplified continuum, the stress interaction is simplified and compression stress and shear stress are uncoupled. Using calculated slab displacements $w$ in normal and $u$ in tangential direction stresses are obtained from

$$\sigma(x) = -\frac{k_\mathrm{n}}{b}w(x), \qquad \tau(x) = \frac{k_\mathrm{t}}{b}u(x). \tag{20}$$

Weak interface approaches are well established in structural analysis (Krenk, 1992; Selvadurai, 1979), in particular when strong elastic contrasts are present. Stress solutions are generally of good quality. However, highly localized effects such as stress singularities cannot be captured. The effect of this limitation is discussed in part II of this work.

## 2.4 Fracture mechanical quantities

Fracture mechanics (Broberg, 1999; Anderson, 2017) is concerned with the behavior of cracks in continua. Propagation of existing cracks can be described in terms of energy considerations which allow for an assessment of the stability of cracks. If the change in total potential energy $\mathrm{d}\Pi$ with an infinitesimal crack advance $\mathrm{d}a$ (denoted as *differential* energy release rate) equals the fracture toughness

$$\mathcal{G} = -\frac{\mathrm{d}\Pi}{b\,\mathrm{d}a} = \mathcal{G}_\mathrm{c}, \tag{21}$$

the crack will grow. This fundamental condition is called Griffith criterion. The fracture toughness describes the energy required for formation of a new crack surface of unit area. It comprises surface energy as well as dissipative energy terms. The latter render fracture processes irreversible. The fracture toughness is defined within the continuum framework. That is, it efficiently covers all local failure mechanisms on the micro-scale – even for complex and heterogeneous materials.[1]

Within the framework of weak interface models the energy release rate corresponding to infinitesimal crack growth can be obtained from the local strain energy (cf. Krenk, 1992). For mode I and mode II contributions to the energy release rate

$$\mathcal{G}_\mathrm{I} = \frac{k_\mathrm{n}}{2b}w(a)^2, \qquad \mathcal{G}_\mathrm{II} = \frac{k_\mathrm{t}}{2b}u(a)^2 \tag{22}$$

hold. Here, $w(a)$ and $u(a)$ correspond to displacements at the crack tip. Again, this simplified framework typically provides good results but it does not capture certain effects such as the energy release rate of vanishing crack lengths (cf. discussion in section 4).

We distinguish different crack opening modes. Mode I loading is a crack opening mode normal to the crack faces. Strictly speaking this comprises only symmetric deformations which typically does not hold for cracks along material discontinuities

---

[1]In linear elastic materials the Griffith criterion may be reformulated using stress intensity factors $K$ which describe the asymptotic crack tip stress field. Both, the critical energy release rate $\mathcal{G}_\mathrm{c}$ and the critical stress intensity factor $K_\mathrm{c}$ are denoted fracture toughness. The units of these two quantities are different but can be converted using the material's stiffness.





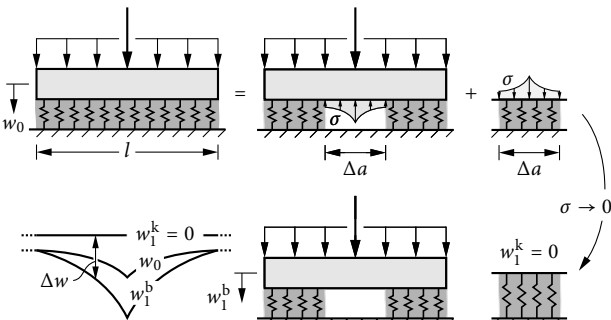

**Figure 4.** Graphical representation of the mode I crack opening integral. An uncracked snowpack can be represented by removing a part of the weak layer and applying virtual stresses to both the slab and the removed weak layer segment such that they are deformed as in the original configuration ($w_0$). Reducing the virtual stresses to zero quasi-statically increases the beam deformation ($w_1^{\mathrm{b}}$), relaxes the weak layer ($w_1^{\mathrm{k}}$) and finally yields the cracked configuration. The work done by the virtual stresses (integral of $\sigma \Delta w$ on $\Delta a$) corresponds to the total potential energy difference between uncracked and cracked configurations.

with different stiffnesses. Mode II and III are shear crack modes with tangential displacements of the two crack faces. In this work only modes I and II are considered. Following the concept of anticracks (Heierli and Zaiser, 2008) we extend the mode I definition to general deformation normal to the crack faces. This includes tearing deformations of the crack faces away from each other but also collapse where crack faces deforming towards each other. Since the micro-structural failure mechanisms

are of course very different for tearing and collapse the magnitude of the associated fracture toughnesses will also be different. Hence, we must distinguish between mode I fracture toughness for tearing $\mathcal{G}_{\mathrm{Ic}}^{+}$ and mode I fracture toughness for collapse $\mathcal{G}_{\mathrm{Ic}}^{-}$.

Loading which causes any combination of the three crack opening modes are called mixed-mode loading. Then mixed-mode failure criteria must be used to account for the different contributions to the total energy release and the required energy to grow a crack under mixed-mode conditions. This will be subject in part II of this work.

In order to extend the scope of fracture mechanics, which is only applicable to infinitesimal crack growth $\mathrm{d}a$, Hashin (1996) considers finite differences in crack length $\Delta a$. Then the so-called *incremental* energy release rate describes the difference in total potential energy $\Delta \Pi$ for a finite crack increment $\Delta a$. The incremental energy release rate can also be obtained integrating the differential energy release rate over the finite crack advance $\Delta a$

$$\overline{\mathcal{G}} = -\frac{\Delta \Pi}{b \Delta a} = \frac{1}{\Delta a} \int_{\Delta a} \mathcal{G} \, \mathrm{d}a. \tag{23}$$

The total differential energy release rate is the sum of contributions from different crack opening modes:

$$\mathcal{G} = \mathcal{G}_{\mathrm{I}} + \mathcal{G}_{\mathrm{II}}. \tag{24}$$

Equivalently, the total incremental energy release rate can be split into contributions from crack opening (mode I) and crack sliding (mode II):

$$\overline{\mathcal{G}} = \overline{\mathcal{G}}_{\mathrm{I}} + \overline{\mathcal{G}}_{\mathrm{II}} = -\frac{\Delta \Pi_{\mathrm{I}}}{b \Delta a} - \frac{\Delta \Pi_{\mathrm{II}}}{b \Delta a}. \tag{25}$$





The total potential energy difference $\Delta\Pi = \Delta\Pi_\mathrm{I} + \Delta\Pi_\mathrm{II}$ between a cracked and an uncracked configuration can be calculated efficiently using the crack opening integral. The change in potential energy corresponds to the work done by stresses on crack flanks when reduced quasi-statically to zero as detailed in Figure 4. The mode I and II contributions amount to

$$\Delta\Pi_\mathrm{I} = -\frac{1}{2}\int\limits_{\Delta A}\sigma\,\Delta w\,\mathrm{d}A = -\frac{b}{2}\int\limits_{\Delta a} -\sigma(x)\,w_1(x)\,\mathrm{d}a,$$

$$\Delta\Pi_\mathrm{II} = -\frac{1}{2}\int\limits_{\Delta A}\tau\,\Delta u\,\mathrm{d}A = -\frac{b}{2}\int\limits_{\Delta a}\tau(x)\,u_1(x)\,\mathrm{d}a. \tag{26}$$

Using the stress to displacement relation (20) obtained in the snowpack model and Eq. (23) yields

$$\overline{\mathcal{G}}_\mathrm{I} = \frac{1}{2b\Delta a}\int\limits_{\frac{l-\Delta a}{2}}^{\frac{l+\Delta a}{2}} k_\mathrm{n}w_1(x)\,w_0(x)\,\mathrm{d}x,$$

$$\overline{\mathcal{G}}_\mathrm{II} = \frac{1}{2b\Delta a}\int\limits_{\frac{l-\Delta a}{2}}^{\frac{l+\Delta a}{2}} k_\mathrm{t}u_1(x)\,u_0(x)\,\mathrm{d}x, \tag{27}$$

where indices 0 and 1 refer to uncracked and cracked configurations, respectively. Computing the incremental energy release rate for a crack of length $\Delta a$ requires knowledge of the deformations of the uncracked and corresponding cracked configurations. These expressions of the incremental energy release rate can be evaluated readily using the displacement solutions of the model presented above.

## 3   Validation of the mechanical model

The present model provides slab displacements, weak layer stresses and energy release rates for cracks within the weak layer as closed-form analytical expressions. In order to validate the model, stresses and energy release rates are compared against detailed finite element analyses (FEA) and existing models. Results of several parametric studies are shown in detail. Further, we compute the fracture toughness corresponding to critical cut lengths in propagation saw tests for a comprehensive set of 93 field experiments provided by Gaume et al. (2017) to investigate the capabilities of the present model.

### 3.1   Reference solution

Reference stress solutions and energy release rates are computed using the plane strain FEA model shown in Figure 5. Like the reference models used by Sigrist and Schweizer (2007) and Habermann et al. (2008) it considers an inclined snowpack consisting of a homogeneous slab and a weak layer. The slab is loaded with a vertical volume gravity load. The weak layer is clamped at its bottom side. Cracks are introduced by removing all weak layer elements on the crack length $a$. The mesh of biquadratic 8-node elements with reduced integration is refined towards stress concentrations. Mesh convergence of the FEA solutions has been controlled. The FEA total energy release rate of a crack of length $\Delta a$ is computed according to

$$\mathcal{G}(\Delta a) = \overline{\mathcal{G}}(\Delta a) + \Delta a\frac{\partial\overline{\mathcal{G}}(\Delta a)}{\partial\Delta a}, \tag{28}$$




**Table 1.** Material properties used throughout the present work.

| Property | Symbol | Value |
|---|---|---|
| Skier weight | $m$ | $80\,\mathrm{kg}$ |
| Slope angle | $\varphi$ | $0\,^\circ$ |
| Slab thickness[*] | $h$ | $40\,\mathrm{cm}$ |
| Weak layer thickness[*] | $t$ | $5\,\mathrm{cm}$ |
| Effective ouf-of-plane ski length | $l_\mathrm{o}$ | $100\,\mathrm{cm}$ |
| Young's modulus slab | $E_\mathrm{slab}$ | $5.23\,\mathrm{MPa}$ |
| Young's modulus weak layer | $E_\mathrm{weak}$ | $0.15\,\mathrm{MPa}$ |
| Poisson's ratio slab & weak layer | $\nu$ | $0.25$ |
| Slab density | $\rho$ | $240\,\mathrm{kg m^{-3}}$ |
| Length of PST block | $l_\mathrm{PST}$ | $120\,\mathrm{cm}$ |

[*]Thicknesses are slope normal.

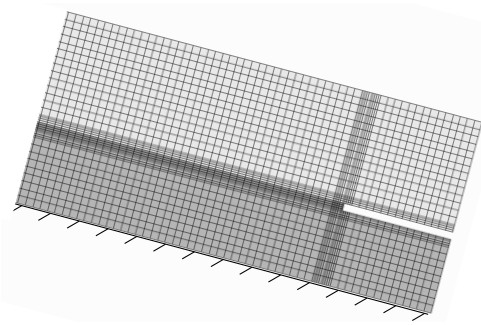

**Figure 5.** Finite element model used for validation. Discretization of a snowpack with slab and weak layer. Cracks are introduced by removing all weak layer elements. Skier loads are applied as vertical concentrated forces. Here, the case of a propagation saw test is shown as an example. The rigid base below the weak layer has a Young's modulus of $E_\mathrm{base} = 10^{12}\,\mathrm{MPa}$.

where the incremental energy release rate $\overline{\mathcal{G}}(\Delta a)$ is determined using Eq. (23). In order to calculate the derivative of $\overline{\mathcal{G}}(\Delta a)$, the incremental energy release rate $\overline{\mathcal{G}}$ is evaluated for four different crack lengths closely around $\Delta a$. The derivative $\overline{\mathcal{G}}(\Delta a)/\partial\Delta a$ is then obtained by differentiating the interpolating cubic spline of the four $\overline{\mathcal{G}}$ values at $\Delta a$.

For the following considerations, the Young's modulus is calculated from density $\rho$ using an empirical power law fit to the data of Scapozza (2004) in plane strain conditions

$$E = \frac{1}{1-\nu^2}\,5.07 \times 10^3 \left(\frac{\rho}{\rho_0}\right)^{5.13} \mathrm{MPa}, \tag{29}$$

with the density of ice $\rho_0 = 917\,\mathrm{kg\,m^{-3}}$.





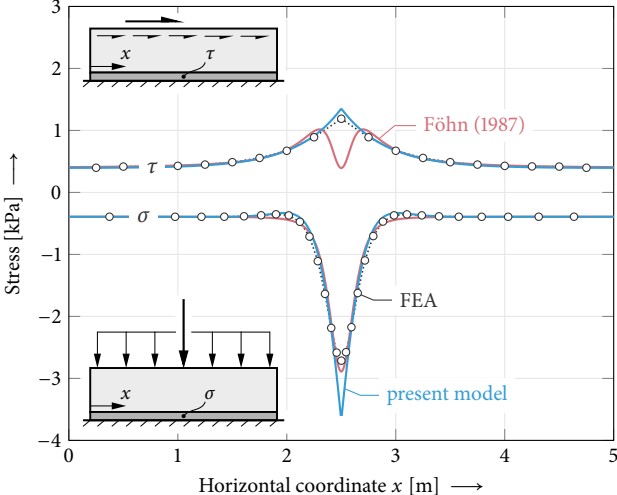

**Figure 6.** Normal and shear stresses owing to pure vertical and pure horizontal combined skier and slab weight loading, respectively. Comparison of present model (blue) and Föhn (1987) model (red) to FEA results (circles). Model parameters other than $h = 20\,\mathrm{cm}$, $t = 1\,\mathrm{cm}$, $E = 5\,\mathrm{MPa}$ and $\rho = 200\,\mathrm{kg\,m^{-3}}$ are chosen as given in Table 1.

## 3.2 Results of validation

Figure 6 shows weak layer compressive and shear stresses owing to pure vertical and pure horizontal combined skier and slab weight loading, respectively. Stresses are shown as calculated using FEA, the solution by Föhn (1987) and the present model. Föhn's solution for a force acting on an elastic halfplane agrees particularly well with the shown FEA results in terms of normal

stress. The present model agrees almost equally good and deviates only in a small region around the load point. Considering shear stress, Föhn provides an exact solution for transverse shear stresses in a homogeneous body. Theses stresses are zero directly below the concentrated force and peak to the left and right of it as seen in Figure 6. Additional shear stresses arise from a horizontal displacement of the beam which also strains the weak layer. Lateral shear stresses originating from this effect peak below the load point. Superimposing both components yields the total weak layer shear stress obtained in FEAs. Note

that lateral shear stresses owing to slope-parallel concentrated force components do not change their sign left and right of the load point as shown in Figure 6. This is in contrast to transverse shear stresses owing to normal concentrated forces which their sign. However, as discussed in section 4 transverse weak layer shear stresses are not accounted for in the present model. It considers only lateral shear where its agreement with FEA results is better than Föhn's transverse shear solution.

     The bridging effect of stiff slabs is studied in Figure 7. The results show that with thicker slabs the local loading of a skier is

transferred on a wider area of the weak layer. This dependence on the slab thickness is very pronounced as the bending stiffness



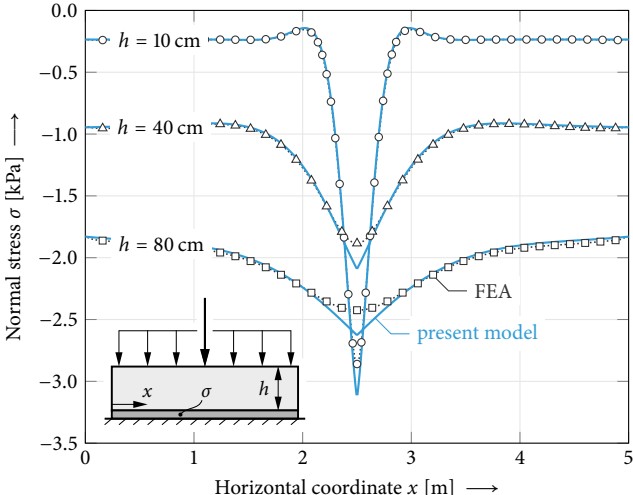

**Figure 7.** Bridging effect evident in weak layer normal stresses depending on the slab thickness. The peak stresses and the width of the skier loaded snow pack changes with different slab thicknesses. The model results (solid lines) are compared against the reference solution (markers $\bigcirc, \triangle, \square$). Model parameters are chosen as given in Table 1.

of the slab increases in cubic dependence with the slab thickness[2]. Because the weight of the slab is proportional to the slab thickness, the stress level distant from the skier increases.

Figure 8 shows the effect of weak layer thickness on normal stresses in the weak layer. Two different weak layer thicknesses are compared. For the thin weak layer we observe strong localization and high peak values while the thick weak layer allows

for point load distribution over a larger area. In comparison to FEA data, a very good agreement of the present model except is evident for either thickness. Very local deviations just below the point load are observed. The Föhn (1987) model is shown for comparison. As it does not account for layering, its normal stress distribution is independent of weak layer thickness.

Let us consider the energy release rate solution. Figure 9 shows the incremental mode I and II energy release rates as a function of slope angle. Besides the implemented Timoshenko kinematics, the limit case of Euler-Bernoulli beam theory is

shown. In the numerical reference model only the total incremental energy release rate is evaluated. The comparison shows that for moderate slope angles the present model provides an excellent prediction of the energy release rate. While the mode I contribution of the energy release rate decreases for higher slope angles, the mode II contribution increases monotonously. Using Euler-Bernoulli beam theory underestimates the mode I contribution and hence the total energy release rate. The mode II energy release rate is a result of tangential displacements and thus unaffected by the choice of beam kinematics.

In Figure 10 a PST experiment is studied. The mode I and mode II energy release rates are shown in dependence of the thickness of the slab above the weak layer. Two different slope inclinations are considered. Energy release rate results are normalized to account for the different orders of magnitude of both crack opening modes. For both angles the mode II contribution

---

[2]The effective rigidity of a Timoshenko beam against vertical deflections is composed of bending stiffness $EI \propto h^3$ and shear stiffness $\kappa GA \propto h$. Hence, its dependence on $h$ is slightly smaller than cubic.





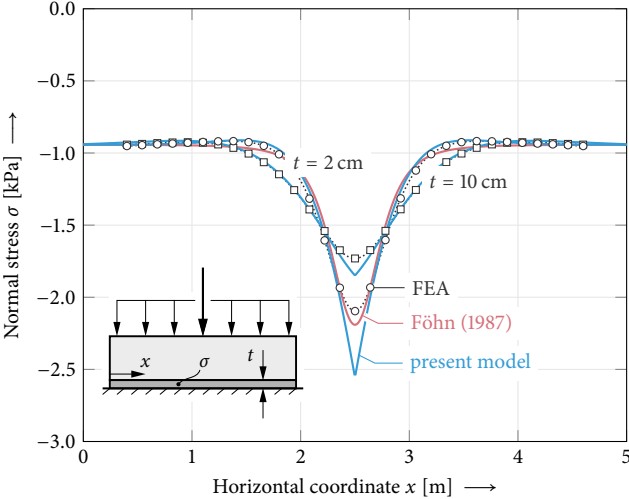

**Figure 8.** Effect of the weak layer thickness on local weak layer normal stresses. The weak layer thickness changes the size of the skier load affected part of the weak layer and also the peak value of the stress. Thicker, more compliant weak layer distribute the load on a larger area and lead to lower peak stresses. The model results (solid lines) are compared against the reference solution (markers $\bigcirc, \square$). Model parameters are chosen as given in Table 1.

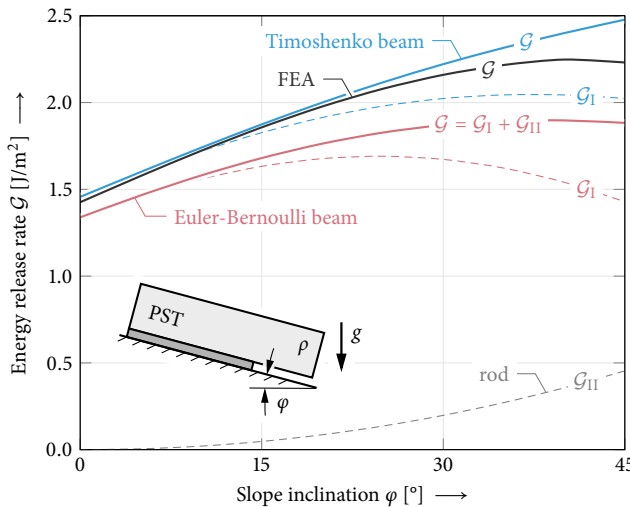

**Figure 9.** Total, mode I and mode II energy release rates in propagation saw tests of different slop angles $\varphi$ with crack lengths $a = 30\,\mathrm{cm}$. Comparison of the present model using Timoshenko beam kinematics accounting shear deformation of the slab and the classical Euler-Bernoulli beam model. The latter is recovered in the limit case of infinite shear stiffness of the beam ($\kappa GA \rightarrow \infty$). Model parameters are chosen as given in Table 1.



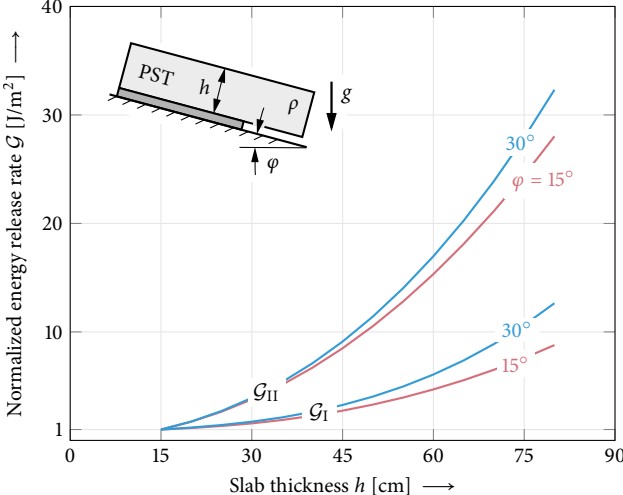

**Figure 10.** Mixed-mode energy release rates of propagation saw test (PST) configurations. The mode I (collapse) and mode II (shear) contributions of the energy release rates of a cut length $a = 30\,\mathrm{cm}$ are shown as a function of the slab thickness. Energy release rates are normalized with the respective initial value at $h = 15\,\mathrm{cm}$: $\mathcal{G}_{\mathrm{I}}^{15°} = 0.805$, $\mathcal{G}_{\mathrm{II}}^{15°} = 0.007$, $\mathcal{G}_{\mathrm{I}}^{30°} = 0.770$, $\mathcal{G}_{\mathrm{II}}^{30°} = 0.028$. Model parameters are chosen as given in Table 1.

has a stronger dependence on the slab thickness than the mode I contribution as they are governed by different deformation mechanisms. Hence, the mode-mixity changes towards a more mode II dominated loading of the PST. No mode III contribution exists when PSTs are cut downsloping and no lateral loading occurs.

Figure 11 compares energy release rates for PSTs in flat terrain computed using FEAs, the anticrack model by Heierli (2008)
and the present model. Different weak layer thicknesses are shown. FEA energy release rates are computed using Eq. (28).
Heierli (2008) models only the unsupported segment of the slab above a failed weak layer. Deformations of the intact weak
layer and the beam segment resting on the intact weak layer are not accounted for. That is, the intact weak layer in Heierli's
model can be envisaged as rigid, i.e., indefinitely thin. For short cracks and thin weak layers Heierli's model agrees well with
FEA results.
PST experiments can be conducted in one of two ways as depicted in Figure 12. The upslope and downslope faces may
be cut slope-normally (Figure 12a) or vertically (Figure 12b) which has to be accounted for when using beam models. A
beam model collapses the slab onto its lower edge resting on smeared springs. Considering vertically cut PSTs, the vertical
gravity load of the slab acts entirely on the beam and is transferred into the weak layer. In the case of slope-normal faces,
the right end of the gravity-loaded slab extends past the right end of its lower edge, i.e., the beam. Hence, the beam is loaded
not only by a distributed load representing gravity loading but also force boundary conditions representing the section forces
and moments resulting from the overhanging mass. The resulting weight force of the overhanging part of the slab amounts
to $F_{+} = \frac{1}{2}\rho g b h^2 \tan(\varphi)$. The corresponding section forces applied at the PST beam model boundary are $N = F_{+} \sin(\varphi)$,
$V = F_{+} \cos(\varphi)$ and $M = \frac{1}{3}F_{+} h \sin(\varphi)$. Please note, this is PST specific. Considering inclined skier-loaded snowpacks with





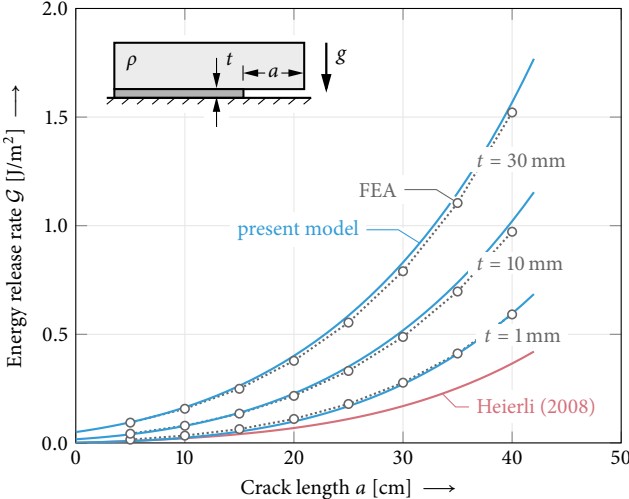

**Figure 11.** Energy release rate $\mathcal{G}$ in flat terrain propagation saw tests (PST). Comparison of present model (blue lines), Heierli (2008) model (red line) and FEA results (gray circles). Both models agree well with FEA data for short crack lengths and stiff (i.e. thin) weak layers. The model by Heierli (2008) cannot reproduce varying weak layer properties providing a lower bound of the energy release rate corresponding to vanishing weak layer thickness. Except for $h = 30\,\mathrm{cm}$ and $\rho = 200\,\mathrm{kgm^{-3}}$ model parameters are chosen as given in Table 1.

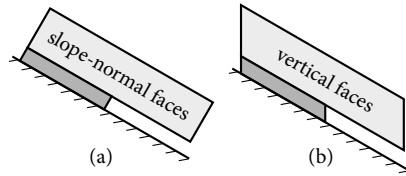

**Figure 12.** Boundary conditions in PST experiments: (a) slope-normal and (b) vertical upslope and downslope faces.

center cracks does not require such boundary conditions because no interaction between boundary effects and skier loading is present.

Figure 13 correlates model predictions for the weak layer fracture toughness to data obtained using detailed FEAs and Eq. (28). The FE model features slope-normal faces such that the corresponding section force boundary conditions discussed above are applied to the beam model. Fracture toughnesses $\mathcal{G}_{\mathrm{c}}^{-}$ are determined from critical cut lengths $a_{\mathrm{c}}$ measured in 93 PST field experiments (Gaume et al., 2017). Measured cut lengths correspond to critical lengths required for crack propagation. No case in the data set showed maximum deflections exceeding the weak layer thickness which would indicate a base-touching slab. According to the fundamental Griffith criterion of fracture mechanics, Eq. (21), the energy release rate corresponding to the critical cut length is the fracture toughness of the weak layer. We will denote it $\mathcal{G}_{\mathrm{c}}^{-}$ to emphasize the difference between the fracture toughness of a collapsing weak layer $\mathcal{G}_{\mathrm{c}}^{-}$ and a tearing fracture toughness $\mathcal{G}_{\mathrm{c}}^{+}$. The difference between the two will be discussed in section 4. Please note, that here only the total energy release rate is considered and no mode-mixity dependence is





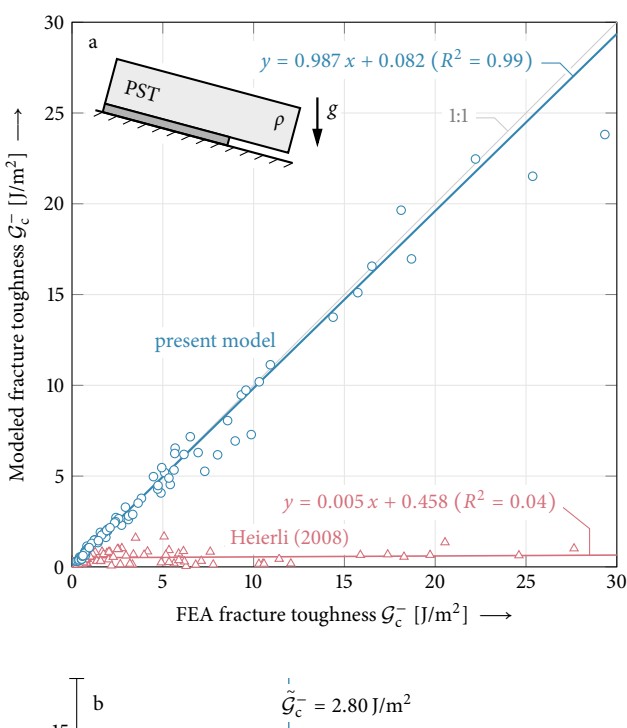

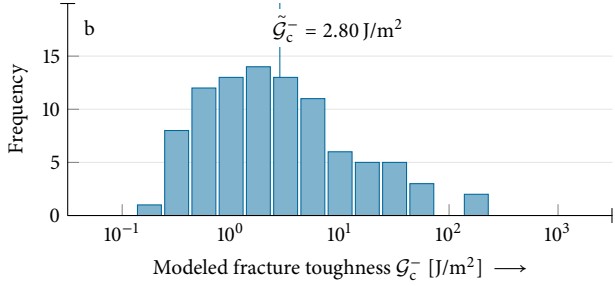

**Figure 13.** Fracture toughness $\mathcal{G}_c^- = \mathcal{G}(a_c)$ determined from 93 field PST experiments reported by Gaume et al. (2017). a) Comparison of model and FEA results with corresponding linear regressions and coefficients of determination $R^2$ for the present model (blue) and the model by Heierli (2008) (red). b) Histogram with logarithmic bin size $10^{0.25}$ and median of the 93 fracture toughnesses calculated using the present model. Data not given in Gaume et al. (2017) are assumed as given in Table 1.





assumed. The 93 PST measurements allow for a model validation against the reference solution and we can obtain actual weak layer fracture toughnesses. For comparison the closed-form analytical expression of Heierli (2008) given by Schweizer et al. (2011) is shown. In the notation of the present work this expression reads

$$
\mathcal{G}_{\mathrm{c}}^{-} = \frac{h}{2E_{\mathrm{slab}}} \left( w_0 + w_1 \frac{a_{\mathrm{c}}}{h} + w_2 \left( \frac{a_{\mathrm{c}}}{h} \right)^2 \right.
$$
$$
\left. + w_3 \left( \frac{a_{\mathrm{c}}}{h} \right)^3 + w_4 \left( \frac{a_{\mathrm{c}}}{h} \right)^4 \right), \tag{30}
$$

with $w_1$ to $w_4$ given in Appendix B.

Predictions of the present model are found within a narrow range around the one-to-one line indicating excellent agreement ($R^2 = 0.99$) with FEAs for this comprehensive set of real world parameters. However, the Heierli solution significantly underestimates the energy release rates as already seen in previous analysis of the influence of the weak layer thickness. The comparison to the FEA reference solution shows poor correlation ($R^2 = 0.04$).

The values of the obtained fracture toughness are within three orders of magnitude. 50% of the values are within one order of magnitude around the median value of $\widetilde{\mathcal{G}}_{\mathrm{c}}^{-} = 2.8 \, \mathrm{Jm}^{-2}$. To individually show most of the data points, Figure 13a only shows a region with values below $30 \, \mathrm{Jm}^{-2}$. As shown in Figure 13b ten data points are above this value. The mean relative error for all 93 data points is 11.1%. Values with $\mathcal{G}_{\mathrm{c}}^{-} > 30 \, \mathrm{Jm}^{-2}$ outside the domain shown in Figure 13a exhibit a slightly increased mean relative error amounting to 12.7%. The relative error of the predicted fracture toughness and the numerical reference model is between $-27.9\%$ and $+45.0\%$. 50% of the values have an error of less than 9.2%. The maximum value of the fracture toughness[3] calculated is $204.4 \, \mathrm{Jm}^{-2}$ with a relative error of $-3.26\%$ in the comparison to the reference model.

The previously given results of the energy release rate as given by the model show *differential* energy release rates $\mathcal{G}$ describing the growth of initial cracks introduced by the saw. In order to study crack initiation it is important to study the *incremental* energy release rate $\overline{\mathcal{G}}$ of cracks of *finite* size, see Eq. (23). In Figure 14 the incremental energy release rate of finite cracks in the weak layer of a skier loaded snowpack are shown. Besides the local load also the weight load of the slab is considered. As an example the effects of slab and weak layer thicknesses are studied. The incremental energy release rate is given as a function of the size of the finite cracks. Of course, the incremental energy release rate increases with crack length. It also increases with slab thickness and with increasing weak layer thickness.

## 4 Discussion

The presented closed-form analytical model of the snowpack contains two different levels of abstraction. The first level is the treatment of the snowpack as a linear-elastic continuum. This is common for models of skier-triggered avalanches (Schweizer and Camponovo, 2001; Schweizer et al., 2003) and agrees with the fact that the brittle failure process of dry-snow slab avalanches occurs within very short time scales (Narita, 1980). Further, (van Herwijnen et al., 2016) have shown a good agreement of displacements measured using particle tracking in field measurements with linear elastic models.

---

[3]This maximum is obtained for the configuration: $h = 81 \, \mathrm{cm}$, $\varphi = 20°$, $\rho = 269 \, \mathrm{kgm}^{-3}$, $t = 2 \, \mathrm{cm}$ and $a_{\mathrm{c}} = 67.1 \, \mathrm{cm}$.





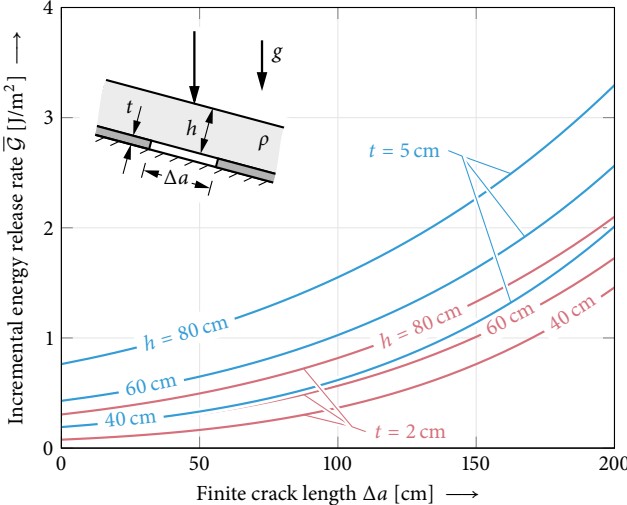

**Figure 14.** Incremental energy release rate of cracks below a skier for three different slab thicknesses. The slab is locally loaded a skier load and by the weight load of the slab. The slab is modelled to have length of 25 times the slab thickness guaranteeing vanishing boundary effects. Model parameters are chosen as given in Table 1.

If it was to study long-term effects within a snowpack like temporal compaction or sintering/metamorphic processes such models are insufficient (Capelli et al., 2018; Mulak and Gaume, 2019). At first glance, the propagation saw test seems governed by the rather slow saw movement. However, the fracture process itself is associated to high strain rates (Reiweger and Schweizer, 2010). The second layer of abstraction is the simplification of heterogeneous media to continua and engineering

structures like elastic foundations or deformable rods and beams. If chosen correctly, accurate representations of the deformation and stress fields within the continua can be recovered with such elements of structural analysis which are well established in civil and mechanical engineering Gross et al. (2014). Using appropriate simplifications numerical models can be avoided and closed-form analytical solutions are obtained. Comparisons to finite element analyses (FEA) (see Figure 9) with continuum elements show that first order shear deformation theory is needed to account for the low shear stiffness of the slab.

Euler-Bernoulli beam theory does not suffice and the present work uses Timoshenko beam theory.

The studies shown in Figures 6 and 8 asses the quality of the stress solution. The present model shows very good agreement with FEA reference solutions with relative errors below 2–5% outside the stress localization domain just below the point load. We exclude this domain from our discussion because considering a localized distributed load instead would resolve this typical problem of concentrated loads and weak interface models. Moreover, the effect of the point load limitation on the

results of failure models is discussed in detail in part II. Stiffer weak layers yield smaller errors. For thicker weak layers stress concentrations below a skier load are less pronounced with larger areas of load transfer (cf. Figure 8). Hence, the peak stresses in the weak layer are reduced with increasing thickness of that layer. This result of the stress solution is in line with the FEA reference solution but cannot be captured by the Föhn (1987) solution which makes use of the elastic half-space solution. This





model deficiency is discussed in detail by Gaume et al. (2017). However, the effect of reduced local load is in contrast with observations that thicker weak layers are more likely to fracture and that avalanches are easier to trigger when the weak layer under the slab is thicker (Gaume et al., 2013). The failure analysis presented in part II of this work is able to correctly recover this effect by additionally considering the energy release rate as a requirement for failure. Figure 7 investigates the bridging

effect. This effect describes the ability to distribute external loads on the slab on a wider area of the weak layer (Thumlert and Jamieson, 2014) as the slab thickness increases. The present model is able to capture the overall stress level due to the increased slab weight as well as the load distribution on a wider area in very good agreement with the numerical reference solution. The load transfer area increases by a factor of approximately five for a change of slab thickness from $h = 10\,\mathrm{cm}$ to $h = 80\,\mathrm{cm}$. Although not shown, the classical solution by Föhn (1987) is capable of rendering this bridging effect correctly. Schweizer

and Jamieson (2003) have introduced a bridging index as the product of slab hardness and thickness. The results of the present analysis show that this is a good estimate but the local stresses decrease overproportionally with the slab thickness. This is due to the bending deformation which is controlled by the bending stiffness of the slab, which in turn is proportional to $h^3$. The effect of the elastic contrast of the stiffnesses of weak layer and slab on this bridging behavior has been addressed by Monti et al. (2015). They proposed to scale the effective slab thickness with the cubic root of the elastic contrast of slab and weak

layer. The bridging effect is very important and explains why thicker slabs can sustain higher loads before weak layer failure occurs (Schweizer and Camponovo, 2001). However, propagation of cracks is more likely below thicker slabs (van Herwijnen and Jamieson, 2007). This change of parameter effect is addressed in the failure model presented in the second part of this work.

The energy release rate of cracks within the weak layer is studied in detail in the validation studies shown in Figures 9 to

13. The effect of slope angle (Figure 9) on the energy release rates shows that both mode I and mode II contributions strongly depend on the slope angle. While the angle dependence of stress solution mainly stems from the split in normal and tangential force contributions, the angle dependence of the energy release rate also depends on the deformations of the slab above the cracked weak layer (cf. Figure 4). Therefore, the angle dependence of simplified models like McClung (1981) or Heierli et al. (2011) is insufficient. The present model is in very good agreement with the numerical reference solution that is carefully set

up to identify these important fracture mechanical quantities.

The effect of the slab thickness on the energy release rate obtained for PST experiments is shown in Figure 10. As mode I and mode II are associated with very different failure modes of collapse and shear failure of the weak layer, the corresponding energy release rates can be of different magnitudes and at the same time both be relevant for the mixed-mode failure. This has been studied in the comprehensive experimental work of Birkeland et al. (2019). They performed PST experiments in which

the slab thickness was changed in order to assess the change of failure behavior at hand of the measured critical cut lengths. The results of the present model show different ratios of mode I and II contributions when the slab thickness changes. This may allow for identifying a comprehensive mixed-mode fracture envelope using these extended PST experiments. The present model could also help to identify relevant geometric parameter ranges to be studied within experimental campaign.

As pointed out by Reiweger and Schweizer (2010) up to 90% of the total deformation of snowpacks is concentrated in the

weak layer because of its considerably low stiffness. It is evident that in order to capture snowpack deformations adequately, a





mechanical model must consider the weak layer. Increasing the thickness of the weak layer while keeping all other geometry and material properties constant increases its compliance. That is, Heierli's assumption of rigidity leads to increasing discrepancies as the weak layer thickness increases. This is reflected in Figure 11. The importance of considering the compliance of the weak layer below the slab is also evident in Figure 13. The thickness of the weak layer (which influences its compliance

linearly) has a pronounced effect on the energy release rate obtained for these cases. The model by Heierli and Zaiser (2008) which neglects the compliance of the weak layer and renders it as rigid is shown in comparison. It is a lower bound of the energy release rate for vanishing weak layer thickness. van Herwijnen et al. (2016) also showed the insufficient approximation for longer cracks and resorted to use a numerical (FEM) correction factor. The energy release rate predicted by the present model which accounts for weak layer deformation is in good agreement with the numerical reference. Figure 9 indicates that

with higher slope angles above 35° the agreement of present solution and reference solution would decrease by a few percent. However, the overall effect of the weak layer thickness (Figure 11) remains the same.

     The analysis of 93 PST data points (Figure 13) provides insight into the analysis of the energy release rate. The 93 data points show a very wide span of the input parameters slab density (factor $4.6$), slab thickness (factor $3.9$), weak layer thickness (factor $44$) and slope angle (from 0° to 44°). They provide a realistic overview on possible configurations along with corresponding

measurements of the critical cut length. Applying the model to these data points now not only provides a comprehensive overview on the models capabilities by comparing it to the numerical reference model but also provides insight into the fracture toughness of a multitude of weak layers. The comparison to the numerical reference model shows that the present model provides a good prediction of the energy release rate for most of the configurations. Only eight ($8.6\%$ of the data points) data points deviate by more than $25\%$ from the numerical reference. Since the energy release rate has a quadratic dependence

on (skier) loading this constitutes an error in a crack propagation analyses below $12\%$. Because Heierli's model neglects weak layer deformations, it underestimates the fracture toughness significantly (see Figure 11). It neither shows a satisfactory slope of the linear regression nor a reasonable coefficient of determination $R^2$. The fracture toughness values that are obtained in the analysis of these 93 data points spans over several decades. However, $75\%$ of the values are between $\mathcal{G}_c^- = 0.3\,\mathrm{Jm}^{-2}$ and $9.7\,\mathrm{Jm}^{-2}$ with a median of $2.0\,\mathrm{Jm}^{-2}$. This range of values reflects the heterogenity of weak layer structures of persistent

weak layers. Complex and divers microstructures are observed for weak layers of faceted crystals or buried surface hoar (Hagenmuller et al., 2014; Schweizer et al., 2003). Possible thickness dependence or layering effects are neither considered in the present nor in the numerical reference model. However, because of the quadratic load dependence of the energy release rate the range of orders of magnitudes of the corresponding crack driving forces are halved when investigating crack initiation and crack propagation.

In the framework of continuum mechanics the PST must be considered a fracture mechanical experiment aimed at identifying the fracture toughness. The fracture toughness is the energy required to form a new surface on the idealized plane of fracture. This energy comprises the dissipative processes at the micro-scale. These processes are much different between crack growth under tension and under compression with a collapse of the weak layer. In the latter case local dissipative damage processes are much more pronounced, leading to a significantly higher value of the (effective) fracture toughness in compression $\mathcal{G}_c^-$

than in tension $\mathcal{G}_c^+$. This has been in observed in lab experiments of cracks on glass foams leading to critical energy release



rates two orders of magnitude higher in compression than in tension Heierli et al. (2012)[4]. For such porous materials with thin micro-structure local stability failure also contributes significantly to the local damage processes. Sigrist (2006) and McClung (2007) show that typical values of the fracture toughness in tension are between $10^{-1}\,\mathrm{Jm^{-2}}$ and $10^{0}\,\mathrm{Jm^{-2}}$, which provides the same relation of tensile and compression fracture toughness as in the experiments by Heierli et al. (2012). The tensile fracture

toughness of ice ($\mathcal{G}_c^+$) is at the order of $10^{0}\,\mathrm{Jm^{-2}}$ (Dempsey, 1991). This is, of course, larger than the fracture toughness of snow in tension. However, the fracture toughness for the present case of mode I in compression may lie well above this value as the corresponding failure process of collapse dissipates significantly more energy than a simple tensile bond breaking. Compared to temperature and time-driven transformation processes of snow, PSTs represent rather fast experiments with little or no impact of viscous effect. Hence, material properties determined from PSTs may be used for analyses of skier-triggered

snowpack instability which is associated to fast loading and high strain rates.

As pointed out by Gaume et al. (2017), fracture parameters obtained from experimental results are always linked to the failure mode that is considered in the model employed to evaluate the data. Hence, fracture energies obtained under the assumption of a rigid weak layer (Heierli et al., 2010) or in pure shear models (McClung, 1979) will always be different from models that take into account weak deformation and mixed-mode fracture. If the failure mode is modeled insufficiently, the obtained failure

parameters are not true material parameters and hence not consistent.

Figure 14 shows the result of incremental energy release rate of cracks of finite size (as opposed to infinitesimal crack growth). These effects are very similar to those the previously discussed case of the differential energy release rate. When differential energy release rates can be calculated as a function of the crack length, incremental energy release rates can always be obtained from integration. The integration is highly efficient using closed-form analytical solutions and established

(numerical) integration schemes. However, computational effort can be reduced even further employing the crack opening integral. While integrating the differential energy release rate requires its evaluation for several crack lengths, the crack opening integral only needs data from the uncracked configuration and the configuration with the final crack length. It computes the difference in total potential energy between an uncracked state and a configuration with finite crack from the work done by stresses as the crack opens. Similar to differential energy release rates, incremental energy release rates increase with increasing

crack length (Figure 14). More importantly, they also increase with both weak layer thickness and slab thickness. Maximum deflections of the slabs considered in Figure 14 are smaller than half the weak layer thickness even for the longest cracks. At first glance, Figures 7 and 14 demonstrate a contradiction: Thicker slabs distribute loads more evenly over the weak layer, reduce local stress peaks and should be more difficult to trigger. However, they release more energy favoring crack propagation. A failure criterion using both stress and incremental energy release rates to asses the nucleation of finite cracks within the weak

layer resolving this contradiction will be introduced in part II of this work.

The present model does not account for layering of the slab above the weak layer. It is well known that the layering can have a significant effect (e.g., Schweizer, 1993; van Herwijnen and Jamieson, 2007; Habermann et al., 2008; Reuter et al., 2015). It influences the load distribution, the fracture process of the weak layer and also the bridging effect. Monti et al. (2015) have

---

[4]In their study Heierli et al. (2012) use critical stress intensity factors, which relate to the critical energy release rate through $\mathcal{G} = K_I^2/E$, and report $K_c^-$ one order of magnitude larger than $K_c^+$

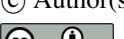



proposed a stability index based on the Föhn solution that accounts for layering by using adapted values of the thickness and the stiffness of the slab. In the present work, rigid base layers are assumed. However, as shown by Jones et al. (2006) and Habermann et al. (2008) substratum elasticity is not negligible. The elastic foundation should not only account for weak layer compliance but also for deformable base layers. Future research is required to make use of these considerations in a refined
version of the present model to combine the capabilities.

Penetration depths leading to forces acting not on top of the slab but well within the slab are not considered at this point. To understand loading scenarios where the penetration depth is high, i.e. because of a soft top layer of snow or a highly localized load (recreationist by foot or a snowmobile) further modeling and experimental effort is needed.

In this work, interaction of shear and compression is limited. Slab extension causes only tangential and slab deflection causes
only normal weak layer deformations. This modeling strategy can be pictured as weak layer springs attached to the mid-surface of the beam. With this simplification the governing system of equations is uncoupled and can be solved independently for tangential and normal displacements, respectively. Of course, the weak layer does not interact with the mid-surface of the slab but with its bottom side. Tangential displacements on the slab bottom side are caused by both slab extension and the rotation of beam cross sections which in turn depends on the slab deflection. Hence, the rotation of beam cross sections couples tangential
and normal displacements. This will increase the solution effort but simultaneously provide improved accuracy, in particular concerning weak layer shear stresses and mode II energy release rates (see, e.g., Figure 6).

## 5   Conclusions

By considering a deformable weak layer the present work provides a simple but comprehensive closed-form analytical model for snowpack deformations, weak layer stresses and energy release rates of cracks within weak layers:

1.  The model applies to skier loaded slopes as well as PST experiments.

2.  Providing closed-form solutions the present analysis framework is highly efficient and evaluates in real-time.

3.  Comparisons with numerical reference solutions indicate good agreement of weak layer stress and energy release rates.

4.  The model renders physical effects such as the bridging effect of thick slabs, the influence of slope inclination and most importantly the impact of weak layer compliance. Evaluating a comprehensive set of 93 PSTs fracture toughnesses of a
multitude of weak layers are analyzed and discussed.

5.  Providing weak layer stress and energy release rates of cracks within the weak layer the present framework allows for evaluating comprehensive criteria for skier-triggered crack nucleation and crack propagation in the weak layer which will be discussed in part II of this two part work.





## Appendix A: Derivation of governing equations

The set of differential equations governing the present problem can be derived from the equilibrium of forces and moments of a beam element and elasticity laws for the bending moment $M$, the transverse shear force $Q$ and the normal force $N$. The latter read (Timoshenko and Goodier, 1951)

$$N = EAu', \tag{A1}$$

$$Q = \kappa GA\left(w' + \psi\right), \tag{A2}$$

$$M = EI\psi', \tag{A3}$$

where $A = hb$ is the snow slab cross section, $\kappa = 5/6$ is the shear correction factor for rectangular cross sections and $I = bh^3/12$ the moment of inertia with respect to the $y$-axis. In the limit $\kappa GA \to \infty$ we obtain the classical Euler-Bernoulli beam theory. Equilibrium of forces and moments of a beam element yields

$$N' = k_\mathrm{t}u - q_\mathrm{t}, \tag{A4}$$

$$Q' = k_\mathrm{n}w - q_\mathrm{n}, \tag{A5}$$

$$M' = Q, \tag{A6}$$

where $k_\mathrm{t}u$ and $k_\mathrm{n}w$ are distributed reaction forces acting on the beam which originate from the elastic foundation. Therefore, they depend on the normal and tangential displacements along the the top of the weak layer and its normal and lateral shear stiffnesses $k_\mathrm{n}$ and $k_\mathrm{t}$.

Preliminary studies showed that two-parameter foundation models like Pasternak or Vlazow foundations (Selvadurai, 1979), which consider additional stiffnesses such as a transverse foundation shear stiffness, do not have a significant effect on the outcome of the analysis.

The ODE governing horizontal displacements, i.e., the deformations of an elastically bedded rod is obtained by inserting the derivative of (A1) into (A4). The following rearrangements are necessary to obtain differential equations of the deflection $w$ and the curvature $\psi$ of a Timoshenko beam on an elastic foundation: Plugging the derivative of (A2) into (A5) yields

$$\kappa GA\left(w'' + \psi'\right) = k_\mathrm{n}w - q_\mathrm{n}. \tag{A7}$$

Adding $0 = EIw'' - EIw''$ to the balance of moments (A6), differentiating twice and plugging the result into the first derivative of (A6) yields

$$Q' = M'' = EI\left(w'''' + \psi'''\right) - EIw''''. \tag{A8}$$

Now we use the third derivative of the elasticity law of shear deformations (A2) to substitute the first term of the right hand side and obtain

$$Q' = \frac{EI}{\kappa GA}Q''' + EIw''''. \tag{A9}$$





Substituting $Q'$ using (A5) and $Q'''$ using the second derivative of (A5) yields the ODE of the beam deflection:

$$q_\mathrm{n} = EIw'''' - \frac{EIk_\mathrm{n}}{\kappa GA}w'' + k_\mathrm{n}w. \tag{A10}$$

In order to obtain the ODE of the rotation of normals to the beam mid-surfce we insert (A2) and the derivative of (A3) into (A6) which yields

$$EI\psi'' = \kappa GA(w'' + \psi). \tag{A11}$$

Plugging the derivative of (A7) into (A11) and rearranging for $\psi$ yields

$$\psi = \left( \frac{EI\,k_\mathrm{n}}{(\kappa GA)^2} - 1 \right) w' - \frac{EI}{\kappa GA}w'''. \tag{A12}$$

## Appendix B: Constants of Heierli's solution

With normal and shear loading $\bar{q}_\mathrm{n} = -\rho gh\cos\varphi$ and $\bar{q}_\mathrm{t} = \rho gh\sin\varphi$, respectively, the constants $w_1$ to $w_4$ of Eq. (30) read

$$w_1 = \frac{3\eta}{4}\bar{q}_\mathrm{t}^2, \tag{B1}$$

$$w_2 = \left( \pi\gamma + \frac{3\eta}{2} \right) \bar{q}_\mathrm{t}^2 + 3\eta^2\bar{q}_\mathrm{n}\bar{q}_\mathrm{t} + \pi\gamma\bar{q}_\mathrm{n}^2, \tag{B2}$$

$$w_3 = 3\eta\bar{q}_\mathrm{n}^2, \tag{B3}$$

$$w_4 = 3\bar{q}_\mathrm{n}^2, \tag{B4}$$

in the notation of the present work where $\gamma \approx 1$ and $\eta = \sqrt{4\,(1+\nu)\,/5}$ are constants.

*Code availability.*  The code of both the modeling framework in part I and the mixed-mode failure criterion based on this framework will be published in an online GitHub repository for public access.

*Author contributions.*  PLR and PW designed, conceived and developed the model. PW drafted the paper and PLR carried out the numerical reference analysis. Both authors contributed equally to the the presented model and the final version of the paper.

*Competing interests.*  The authors declare that they have no conflict of interest.

*Acknowledgements.*  We thank Johan Gaume and Alec van Herwijnen for sharing their PST data set and for valuable discussions on snow slab models and characterization tests. We want to thank Ned Bair, Karl Birkeland and Bastian Bergfeld for fruitful discussions on snow



physics, model assumptions and snowpack stability experiments. We thank Wilfried Becker for his support and Vera Lübke for studying the feasibility of different modeling approaches. We acknowledge support by the German Research Foundation and the Open Access Publishing Fund of Technische Universität Darmstadt.



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
