# Peer review of "Modeling snow slab avalanches caused by weak layer failure – Part I: Slabs on compliant and collapsible weak layers"

_The Cryosphere, 2019_

## Referee Comment (RC1) · Anonymous Referee #1 · 18 Jun 2019

Modeling snow avalanches caused by weak layer failure Part I: slabs on compliant and collapsible weak layers

Author: P.L. Rosendahl, and P. Weissgraeber

General comments

This manuscript deals with the failure modeling within a snow mantel including a weak layer. This topic is of importance in snow engineering, as just a little is known today about how failure occurs and which parameters control the failure propagation within a snowpack. The authors have developed an original, analytical approach, based on elasticity equations and crack propagation. The basic idea is that the weak layer is

prone to collapse, triggering a cracking mechanism within the subsequent layers. Even though this approach is appealing as it is quite easy-to-use, some remarks should be raised. In particular, the assumption that the computation can be run starting from an elastic state is questionable. What does elasticity mean in snow? Snow is first and foremost a (cohesive) granular material, in which the elasticity domain is very limited. Plastic dissipation develops early. Additional remarks are reported below.

In conclusion, I consider that this manuscript should be revised before being considered for publication in The Cryosphere.

Specific comments

Introduction section Line18 "If the conditions allow for crack propagation,..." Could the authors be more specific?

Mechanical modeling Failure initiation and propagation is a truly 3D problem. How can this approach be extended in 3D conditions?

Boundary conditions are supposed to play a very important role in failure propagation. How can this aspect be accounted for?

FEM computations make it possible to estimate both strain and stress within the snow pack, even though inhomogeneous conditions are considered. Thus, what is the interest of applying a simplified approach? In addition, more complex mechanical constitutive relations can be used in FEM (visco-plastic models, etc.). Could the authors develop a little bit more along this line?

Weak layer collapse under the snowpack can be regarded as the triggering event preceding snowpack failure. How this collapse can reasonably be considered in the present approach?

Equation 29 The Young modulus is expressed as a function of the relative density of snow. Does the temperature play no role and can it be ignored? Is it the expression of the Young modulus, referring to a truly elastic behavior, or does this term refer to

the compressibility of a given snow specimen, irrespective of the behavior (elastic or anelastic)?

Finally, more recent numerical tools exist to deal with the mechanical behavior of snowpack, including the detection of failure. DEM (discrete element method) approaches stand probably as a convenient and promising way. Also, micro-mechanically based constitutive approaches should be mentioned. It is regrettable that the authors have completely ignored this part of the state-of-the art. This should be considered in the revised version.

---

## Referee Comment (RC2) · Michael Zaiser (Referee) · 20 Jun 2019

The paper revisits the mechanical model for brittle failure of a weak layer under compressive and mixed loads as proposed by Heierli and co-workers. The model shares the conceptual foundation of that model and its limitations (brittle vs quasi –brittle behavior of snow, use of a purely energy based criterion for fracture propagation which does not account for the actual stress state at the crack tip). Within these limitations, the work, which consists of analytical considerations validated by FEM calculations, is solidly done. The main novelty introduced by the authors consists in the inclusion of weak layer elasticity, which is considered in terms of a spring model ('Winkler foundation') commonly used in adhesion mechanics. Methodologically this is fine.

The mathematical results are reasonable. First it is clear that accounting for elastic energy stored in the weak layer is bound to increase the energy release rate in comparison with the results of Heierli et al. who treat the interface layer as stiff. Second, an elementary estimate shows that the relative energy stored in the weak layer and in the slab under a fixed gravitational load of order $\sigma$ is proportional to

$$E \propto h_i \sigma \epsilon_i = \sigma^2(h_i/E_i)$$

where $\epsilon_i$ is a measure of the characteristic strain in layer i (slab or weak layer), $h_i$ is the respective layer thickness and $E_i$ is the elastic modulus of the layer. This gives also a rough measure of the respective contribution to the energy release rate. In simple words, weak layer elasticity is relevant if the layer is thick and elastically soft.

Thus, it is in the nature of the case put forward by the authors that they are considering effects that are relevant for thick soft weak layers, and much less so for layers that are thin or elastically similar to the slab. This is borne out by the parameters compiled in Table 1: The weak layer thickness is assumed 5 cm, (thick), the slab thickness is 40 cm (thin), the elastic modulus of the slab is 35 times higher than of the weak layer.

A quantitative comparison with results of the Heierli model is given in Figure 11. For a WL thickness of 3cm, and the other parameters as in Table 1, my simple estimate shows that the energy stored in the WL is about 3 times higher than for the slab, accordingly the overall energy release rate in Figure 11 for that thickness is about 4 times higher than in the Heierli model. The results are thus what one expects. Good.

Thus, the authors show that for some cases the influence of the weak layer on energy release rate may be significant. They in my opinion seriously over-state their case when they imply, by their choice of parameters, that it is always predominant.

(i) First let us note that the elastic modluus values used by the authors are dubious in absolute terms. Elastic moduli of snow can be inferred computationally from FEM on

snow microstructures determined by micro-CT, and these calculations can be experimentally validated based on elastic wave propagation data, see [1] Gerling, B., Löwe, H., van Herwijnen, A. (2017). Measuring the elastic modulus of snow. Geophysical Research Letters, 44, 11,088–11,096 and [2] Koechle and Schneebeli, Journal of Glaciology, Vol. 60, No. 222, 2014. The authors should address the discrepancy between those data and the elastic moduli used in their computations.

(ii) Irrespective of absolute numbers, snow elastic moduli are highly density dependent, scaling in approximate proportion with the fourth power of density [1] and following the same density vs modulus curve for both weak layers and bulk snow [2]. Thus, differences in weak layer and slab density of a factor 2 can indeed account for significant differences in modulus. Nevertheless the assumptions of Table 1 seem excessive - to explain the modulus ratio of a factor of 35 assumed by the authors, the weak layer density would need to be around 100 $kgm^{-3}$. The authors should provide evidence that such huge density differencess between slab and weak layers are indeed common, e.g. in experimental snow density profiles (BTW I have a few counter examples at hand).

Also one may note that weak layer density relates to collapse height. Under the reasonable assumption that the weak layer compacts, during collapse, at least to the density of the overlying slab, a layer of thickness 5cm compacting from 100$kgm^{-3}$ to 240$kgm^2$ would entail a collapse height of about 3cm which appears excessive compared with collapse heights observed in field experiments (propagation saw tests) published in the literature.

In summary, it should be clearly explained by the authors that the difference between the present model and the previous model of Heierli et al is contingent on a very significant modulus (density) difference between slab and weak layer, and the authors should discuss, from a snow science perspective and providing appropriate evidence, under which circumstances such modulus/density differences are to be expected. This would help to put the results into context and to illustrate their practical relevance. They

should explicitly relate their parameter assumptions to field data e.g. on propagation saw tests and demonstrate that they are reasonable in view of established relationships between density, modulus, and in view of observed weak layer thicknesses and collapse heights. If the results are thus put into perspective, I think the paper should be published since it sheds light on an aspect of weak layer collapse which, while in real world situations most probably not as dominant as the authors try to suggest, may in some circumstances be of relevance for the interpretation of propagation saw test data and snow stability in general.

---

## Author Comment (AC2) · 12 Jul 2019

The anonymous referee has detailed his or her view on the modeling framework itself and addressed other material and structural modeling concepts such as plasticity as well as a three-dimensional modeling approach to better capture the behavior of the snowpack.

In our detailed point-by-point response below we explain that we have chosen a linear elastic framework due to the nature of brittle onset of anticracks and that we are aware of the simplifications. We will further improve their discussion within the manuscript. The two-dimensional modeling framework was chosen because it provides direct in-

sight into the effects of many important parameters. Both assumptions are common for studies of dry-slab avalanche release.

We thank the anonymous referee for the detailed comments on our manuscript. We have addressed all comments below and have modified the manuscript according to the reviewer's concerns.

**General reviewer comments**

This manuscript deals with the failure modeling within a snow mantel including a weak layer. This topic is of importance in snow engineering, as just a little is known today about how failure occurs and which parameters control the failure propagation within a snowpack. The authors have developed an original, analytical approach, based on elasticity equations and crack propagation. The basic idea is that the weak layer is prone to collapse, triggering a cracking mechanism within the subsequent layers. Even though this approach is appealing as it is quite easy-to-use, some remarks should be raised. In particular, the assumption that the computation can be run starting from an elastic state is questionable. What does elasticity mean in snow? Snow is first and foremost a (cohesive) granular material, in which the elasticity domain is very limited. Plastic dissipation develops early. Additional remarks are reported below.

In conclusion, I consider that this manuscript should be revised before being considered for publication in The Cryosphere.

Presuming linear elasticity is the most fundamental assumption of the present work. This assumption is common for models of skier-triggered avalanches. See, for instance [1, 2, 3, 4, 5, 6, 7, 8], to name only a few. The assumption agrees with the observation that dry-snow slab avalanche release occurs within short time scales [9]. We would describe snow rather as visco-elastic than plastic. That is, dissipative effects are of minor importance in the presence of rapid deformations caused by skiers. Further, van Herwijnen et al. [10] have shown a good agreement of displacements measured using

particle tracking with linear elastic models. We always prefer to use models that use a minimum level of complexity for capturing and understanding physical mechanisms at play. For instance, closed-form analytical solutions directly elucidate the interplay of different quantities of interest which otherwise may remain hidden behind the colorful curtain of finite element plots.

Please note that while the model presented in this first part of the two-part paper is linear elastic, the failure criterion employed in part II accounts for quasi-brittleness of the material.

**Specific reviewer comments**

Introduction section Line18 "If the conditions allow for crack propagation,. . ." Could the authors be more specific?

In this part of the introduction, it is our intention to point out the importance of distinguishing between crack nucleation and crack propagation. The conditions for crack propagation first and foremost refer to the Griffith criterion, of course. That is, the snowpack must release enough energy to drive the crack. Relevant are in particular weight, stiffness, fracture toughness, slope angle, etc. We have (briefly because it is the abstract) clarified this in the manuscript:

> If the conditions allow for crack propagation *, i.e., if the energy release rate of a growing crack suffices*, a triggered initial defect may extend across slopes and eventually cause the slab to fail and slide.

Mechanical modeling Failure initiation and propagation is a truly 3D problem. How can this approach be extended in 3D conditions?

The approach can be readily extended to a 3D analysis. In the presented work we decided to restrain to a 2D analysis as many physical effects governing the problem
can be studied in 2D. To name only a few, these are for instance, bridging, mode-mixety, slope angle dependence, slab thickness dependence, among others. See, e.g. [11, 12, 13].

An important feature of the present model is considering the weak layer as a so-called weak interface. That is, it is elastic but does not need to be treated as a full continuum but can be modeled using simplified kinematics. This idea can be readily transferred to 3D by replacing the bedded Timoshenko beam with a bedded Reissner-Mindlin plate. In 2D, the weak layer with its simplified kinematics provides a system of differential equations that can be solved in a closed-form analytical manner. The 3D case will yield a system of coupled partial differential equations whose solution is more involved.

Boundary conditions are supposed to play a very important role in failure propagation. How can this aspect be accounted for?

The present work considers displacement differential equations. Boundary conditions for this kind of boundary value problem are given in terms of vertical beam deflections, horizontal beam displacements, rotations of the beam cross section, normal and shear section forces as well as section moments. For the propagation saw test, all sides of the isolated snowpack column are free ends. Hence, the corresponding appropriate boundary conditions are vanishing section forces and moments, which we employed in the present work. For skier loaded slopes, we considered free ends with vanishing section forces and moments, as well. This is convenient because their implementation is straightforward, and the analytical framework allows for considering large enough domains (without increasing the computational effort) so that all boundary effects have decayed. If crack propagation and long cracks are to be considered, we could instead use so-called boundary conditions at infinity representing a domain extending to infinity.

FEM computations make it possible to estimate both strain and stress within the snow pack, even though inhomogeneous conditions are considered. Thus, what is the interest of applying a simplified approach? In addition, more complex mechanical con-

*stitutive relations can be used in FEM (visco-plastic models, etc.). Could the authors develop a little bit more along this line?*

We make use of the elastic framework to obtain a real-time solution that can be evaluated in milliseconds even on a mobile device, e.g., in the field. The efficiency can be exploited for intensive parameter studies as, e.g., uncertainty quantification analyses or inverse parameter identifications (multi-parameter fits) in which easily 10.000 to 100.000 model evaluations can be required. Of course, FEM is more versatile and allows for arbitrary constitutive laws. These could also be implemented in a closed-form analytical solution but then quite often a numerical solution scheme (as the FEM) is to be preferred.

As mentioned above, major effects that govern the failure process leading to slab-avalanche release are covered by a linear elastic analysis.

*Weak layer collapse under the snowpack can be regarded as the triggering event preceding snowpack failure. How this collapse can reasonably be considered in the present approach?*

To model the collapsed weak layer below the slab and the the consequential loss of load transfer in normal and shear direction we remove the support of the elastic foundation. This changes the structural response (deformation) and provides the energy release rate of weak layer anticracks. The conditions required for the collapse of weak layer (the nucleation of the anticrack) are addressed in part II of this work.

*Equation 29 The Young modulus is expressed as a function of the relative density of snow. Does the temperature play no role and can it be ignored? Is it the expression of the Young modulus, referring to a truly elastic behavior, or does this term refer to the compressibility of a given snow specimen, irrespective of the behavior (elastic or anelastic)?*

The considered experiments contain little to no information of the respective temperatures. We have employed the widely used [7, 13, 14, 15, 16, 17, 18, 19, 20] equation for the dependence of the Young's modulus on the slab density that has been proposed by Scapozza [21]. Of course, it would be interesting to study the effect of the Young's modulus on the slab temperature as well.

As discussed by Köchle and Schneebeli [22], Scapozza's density-modulus relation is based on deformation measurements in the laboratory, i.e., on the actual structural response. Other authors, such as Gerling et al. [23], provide equations for the density-dependence of the Young's modulus derived from FE models of CT scans or acoustic wave propagation measurements. Elastic moduli derived this way correspond to the response of an ideal material. The equations derived by the two different approaches differ in their absolute magnitude but provide the same trend $E \sim \rho^4$, approximately.

Finally, more recent numerical tools exist to deal with the mechanical behavior of snowpack, including the detection of failure. DEM (discrete element method) approaches stand probably as a convenient and promising way. Also, micro-mechanically based constitutive approaches should be mentioned. It is regrettable that the authors have completely ignored this part of the state-of-the art. This should be considered in the revised version.

We agree that the literature review on modeling approaches of part I must be extended by DEM and micromechanics approaches. We have discussed DEM models in the second part when reviewing existing approaches to failure modeling of snowpacks. However, they should not be omitted in part I. In the revised manuscript, we now briefly cover DEM analyses and micromechanical approaches:

> *Considering the complex microstructure of snow, micromechanical models derive the macroscopic constitutive behavior from representative volume elements [24, 25]. Similarly, discrete element models assemble the continuum from individual particles to model the effective structural response [13, 17].*

[1] J. Schweizer and C. Camponovo. The skier's zone of influence in triggering slab avalanches. Annals of Glaciology, 32(1):314–320, 2001.

[2] J. Schweizer, B. Jamieson, and M. Schneebeli. Snow avalanche formation. Reviews of Geophysics, 41(4):1016, 2003.

[3] J. Heierli and M. Zaiser. Failure initiation in snow stratifications containing weak layers: Nucleation of whumpfs and slab avalanches. Cold Regions Science and Technology, 52(3):385–400, 2008.

[4] J. Schweizer, A. van Herwijnen, and B. Reuter. Measurements of weak layer fracture energy. Cold Regions Science and Technology, 69(2-3):139–144, 2011.

[5] J. Gaume, J. Schweizer, A. van Herwijnen, G. Chambon, B. Reuter, N. Eckert, and M. Naaim. Evaluation of slope stability with respect to snowpack spatial variability. Journal of Geophysical Research: Earth Surface, 119(9):1783–1799, 2014.

[6] F. Monti, J. Gaume, A. van Herwijnen, and J. Schweizer. Snow instability evaluation: calculating the skier-induced stress in a multi-layered snowpack. Natural Hazards and Earth System Sciences Discussions, 3(8):4833–4869, 2015.

[7] B. Reuter, J. Schweizer, and A. Van Herwijnen. A process-based approach to estimate point snow instability. Cryosphere, 9(3):837–847, 2015.

[8] J. Gaume and B. Reuter. Assessing snow instability in skier-triggered snow slab avalanches by combining failure initiation and crack propagation. Cold Regions Science and Technology, 144(May):6–15, 2017.

[9] H. Narita. Mechanical behaviour and structure of snow under uniaxial tensile stress. Journal of Glaciology, 26(94):275–282, 1980.

[10] A. van Herwijnen, J. Gaume, E. H. Bair, B. Reuter, K. W. Birkeland, and J. Schweizer. Estimating the effective elastic modulus and specific fracture energy of snowpack layers from field experiments. Journal of Glaciology, 62(236):997–1007, 2016.

[11] S. Thumlert and B. Jamieson. Stress measurements in the snow cover below localized dynamic loads. Cold Regions Science and Technology, 106-107:28–35, 2014.

[12] I. Reiweger, J. Gaume, and J. Schweizer. A new mixed-mode failure criterion for

weak snowpack layers. Geophysical Research Letters, 42(5):1427–1432, 2015.

[13] J. Gaume, A. van Herwijnen, G. Chambon, K. W. Birkeland, and J. Schweizer. Modeling of crack propagation in weak snowpack layers using the discrete ele- ment method. The Cryosphere, 9(5):1915–1932, 2015.

[14] J. Heierli. Anticrack model for slab avalanche release. PhD thesis, Universität Karlsruhe, 2008.

[15] J. Schweizer, B. Reuter, A. van Herwijnen, B. Richter, and J. Gaume. Temporal evolution of crack propagation propensity in snow in relation to slab and weak layer properties. The Cryosphere, 10(6):2637–2653, 2016.

[16] J. Schweizer, B. Reuter, A. van Herwijnen, D. Gauthier, and B. Jamieson. On how the tensile strength of the slab affects crack propagation propensity. In Proceed- ings, International Snow Science Workshop, Banff, pages 164–168, 2014.

[17] J. Gaume, A. van Herwijnen, G. Chambon, N. Wever, and J. Schweizer. Snow fracture in relation to slab avalanche release: critical state for the onset of crack prop- agation. The Cryosphere, 11(1):217–228, 2017.

[18] L. Benedetti, J. Gaume, and J.-T. Fischer. A mechanically-based model of snow slab and weak layer fracture in the Propagation Saw Test. International Journal of Solids and Structures, 2018.

[19] B. Reuter, N. Calonne, and E. Adams. Shear failure of weak snow layers in the first hours after burial. The Cryosphere Discussions, (January):1–17, 2019.

[20] K. W. Birkeland, A. van Herwijnen, B. Reuter, and B. Bergfeld. Temporal changes in the mechanical properties of snow related to crack propagation after loading. Cold Regions Science and Technology, 159:142–152, 2019.

[21] C. Scapozza. Entwicklung eines dichte- und temperaturabhängigen Stoffgeset- zes zur Beschreibung des visko-elastischen Verhaltens von Schnee. PhD thesis, ETH Zürich, 2004.

[22] B. Köchle and M. Schneebeli. Three-dimensional microstructure and numerical calculation of elastic properties of alpine snow with a focus on weak layers. Jour- nal of Glaciology, 60(222):705–713, 2014.

[23] B. Gerling, H. Löwe, and A. van Herwijnen. Measuring the Elastic Modulus of Snow. Geophysical Research Letters, 44:11,088–11,096, 2017.

[24] François Nicot. From constitutive modelling of a snow cover to the design of flexible protective structures Part I—-Mechanical modelling. International Journal of Solids and Structures, 41(11-12):3317–3337, 2004.

[25] F. Nicot. From constitutive modelling of a snow cover to the design of flexible protective structures Part II—-Some numerical aspects. International Journal of Solids and Structures, 41(11-12):3339–3352, 2004.
* * *

---

## Author Comment (AC3) · 8 Aug 2019

Prof. Michael Zaiser has provided a detailed analysis of our model and has raised concerns regarding our choice of the parameters. He argues that for other parameter sets the effect of the weak layer would be less pronounced.

We agree that our weak layer modulus and thickness assumptions can be improved to better agree with field observations. We have, therefore, rerun our calculations with new parameters for the weak layer. The difference to the model of Heierli and Zaiser [1] (that assumes a rigid foundation) is now smaller but still very pronounced. We further clarify other differences between our model and that of Heierli and Zaiser [1].

[Figure]

We thank Prof. Zaiser for the detailed review. We have update the figures corresponding to parameter studies in our manuscript and have extended the discussion considering this input parameter dependence.

**Reviewer comments**

(i) First let us note that the elastic modluus values used by the authors are dubious in absolute terms. Elastic moduli of snow can be inferred computationally from FEM on snow microstructures determined by micro-CT, and these calculations can be experimentally validated based on elastic wave propagation data, see [1] Gerling, B., Löwe, H., van Herwijnen, A. (2017). Measuring the elastic modulus of snow. Geophysical Research Letters, 44, 11,088-11,096 and [2] Koechle and Schneebeli, Journal of Glaciology, Vol. 60, No. 222, 2014. The authors should address the discrepancy between those data and the elastic moduli used in their computations.

We derive the slab's elastic modulus from its density using a power law fit to Scapozza's [2] data (Eq. (29) in our manuscript). We use this equation to compute data that can be compared to the results of many other analyses, also very recent works, that use the same concept [3–11].

The equation provided by Gerling et al. [12] is cross-validated using two different experimental methods, therefore, likely more reliable and should perhaps be used for future works. However, given its large variability ($E_\text{slab}$ between $7\,\text{MPa}$ and $110\,\text{MPa}$ for our assumption of $\rho_\text{slab} = 240\,\text{kg/m}^3$), using Eq. (29) seems reasonable as well. We added the following paragraph to our manuscript:

*Note that Gerling et al. [12] provide a different equation that is cross-validated using two different experimental methods and, therefore, likely more reliable. However, we chose Eq. (29) for comparability with previously published models.*

Our choice of weak layer Young's modulus is based on Köchle and Schneebeli's statement "*the weak layer was, on average, about half as dense as the layers above and below*" [13]. Hence, our assumption of a slab density of $240\,\mathrm{kg/m^3}$ corresponds a weak layer density of approximately $120\,\mathrm{kg/m^3}$. Then, the power law fit to Scapozza's [2] data yields $E_{\mathrm{slab}} = 5.23\,\mathrm{MPa}$ and $E_{\mathrm{weak}} = 0.16\,\mathrm{MPa}$. Although it is not clear whether Eq. (29) is valid for weak layers, this seemed a reasonable first guess.

However, Köchle and Schneebeli also point out that "*the elastic modulus was much higher (on average 20 times) in the layers above and below*" [13]. With $E_{\mathrm{slab}} = 5.23\,\mathrm{MPa}$ this yields $E_{\mathrm{weak}} \approx 0.25\,\mathrm{MPa}$. Such values and corresponding densities are sensible as discussed in the next point. In order to avoid extreme assumptions and to show that our model does not require very large elastic contrasts, we used a weak layer Young's modulus of $E_{\mathrm{weak}} = 0.25\,\mathrm{MPa}$ to recompute relevant parametric studies of the present work and updated our figures accordingly. Doing so did not change any statement qualitatively and all conclusions in the manuscript remain valid. Table 1, all figures and concerned figure captions are updated accordingly. We added a brief statement of our reasoning concerning parameter choices to the manuscript:

*A weak layer Young's modulus of $E_{\mathrm{weak}} = 0.25$ MPa is chosen based on the findings of Köchle and Schneebeli [13] who report an average ratio of weak layer to slab Young's modulus $E_{\mathrm{weak}}/E_{\mathrm{slab}} = 1/20$.*

(ii) Irrespective of absolute numbers, snow elastic moduli are highly density dependent, scaling in approximate proportion with the fourth power of density [1] and following the same density vs modulus curve for both weak layers and bulk snow [2]. Thus, differences in weak layer and slab density of a factor 2 can indeed account for significant differences in modulus. Nevertheless the assumptions of Table 1 seem excessive - to explain the modulus ratio of a factor of 35 assumed by the authors, the weak layer density would need to be around $100\,\mathrm{kg m^{-3}}$. The authors should provide evidence

that such huge density differencess between slab and weak layers are indeed common, e.g. in experimental snow density profiles (BTW I have a few counter examples at hand). Also one may note that weak layer density relates to collapse height. Under the reason- able assumption that the weak layer compacts, during collapse, at least to the density of the overlying slab, a layer of thickness 5cm compacting from $100\,\mathrm{kgm}^{-3}$ to $240\,\mathrm{kgm}^{-3}$ would entail a collapse height of about 3cm which appears excessive compared with collapse heights observed in field experiments (propagation saw tests) published in the literature.

As pointed out in our answer to remark (i), our choice of modulus ratio was indeed based on estimates of density differences (by Köchle and Schneebeli [13]). We cannot say whether such large density differences are common, however, a number of authors points at weak layer densities much lower than the one we initially assumed $(120\,\mathrm{kg/m^3})$. For instance, using two different measurement techniques, Föhn [14] reports densities of surface hoar layers i) between $44$ and $215\,\mathrm{kg/m^3}$ with a mean of $102.5\,\mathrm{kg/m^3}$ and ii) between $75$ and $252\,\mathrm{kg/m^3}$ with a mean of $132.4\,\mathrm{kg/m^3}$. Horton et al. [15] even measure densities as low as $\rho = 30\,\mathrm{kg/m^3}$. As pointed out above, we changed the weak layer Young's modulus in an effort to avoid polarizing assumptions. According to Eq. (29) of our manuscript, the new weak layer modulus corresponds to a weak layer density of $\rho_{\mathrm{weak}} \approx 135\,\mathrm{kg/m^3}$.

Concerning weak layer thickness and collapse heights, we analyzed the data set provided by Gaume et al. [8] and used the data set's mean weak layer thickness $(48\,\mathrm{mm} \approx 50\,\mathrm{mm})$ for our parametric studies. In view of other publications such as the work of Jamieson and Schweizer [16], who report weak layer thicknesses between $2$ and $30\,\mathrm{mm}$, however, we agree that our initial assumption of $5\,\mathrm{cm}$ may seem excessive. Again, in order to avoid extreme assumptions, we have changed the weak layer thickness to $t = 2\,\mathrm{cm}$ for our parametric studies and updated Table 1, figures and corresponding figure captions. We have included the above arguments in our text:

*Assuming Eq. (29) to be applicable to weak layers as well, $E_{\text{weak}} = 0.25$ MPa corresponds to a weak layer density of $\rho_{\text{weak}} \approx 135 \, \text{kg/m}^3$. This agrees with density measurements of surface hoar layers by Föhn [14] who reports densities i) between $44$ and $215 \, \text{kg/m}^3$ with a mean of $102.5 \, \text{kg/m}^3$ and ii) between $75$ and $252 \, \text{kg/m}^3$ with a mean of $132.4 \, \text{kg/m}^3$ using two different measurement techniques.*

*With reference to Jamieson and Schweizer [16] who report weak layer thicknesses between $0.2$ and $3$ cm, we chose $t = 2$ cm. Further parameter choices are summarized in Table 1.*

In summary, it should be clearly explained by the authors that the difference between the present model and the previous model of Heierli et al is contingent on a very significant modulus (density) difference between slab and weak layer, and the authors should discuss, from a snow science perspective and providing appropriate evidence, under which circumstances such modulus/density differences are to be expected. This would help to put the results into context and to illustrate their practical relevance. They should explicitly relate their parameter assumptions to field data e.g. on propagation saw tests and demonstrate that they are reasonable in view of established relationships between density, modulus, and in view of observed weak layer thicknesses and collapse heights. If the results are thus put into perspective, I think the paper should be published since it sheds light on an aspect of weak layer collapse which, while in real world situations most probably not as dominant as the authors try to suggest, may in some circumstances be of relevance for the interpretation of propagation saw test data and snow stability in general.

It is our intention to provide a model of the mechanical behavior of skier loaded slabs on porous and collapsible weak layers. The weak layer's porosity that is required for its collapse implies a certain elastic contrast between slab and weak layer. Slab avalanche release owing to other failure mechanisms such as time-dependent damage accumulation are likely better captured using (gradient) plasticity approaches and such.

We hope that our assumptions and results are put into context with the above arguments and changes to our assumptions. However, we disagree with the statement that "*the difference between the present model and the previous model of Heierli et al is contingent on a very significant modulus (density) difference between slab and weak layer*".

In order to illustrate this, consider the attached Figure 1 where we have recomputed the results shown in Fig. 11 of our manuscript using different weak layer moduli. The graph shows that even when the elastic moduli of weak layer and slab are very similar (ratio 1:2.5 at $E_\text{weak} = 2.0\,\text{MPa}$), there is a significant difference between Heierli's and the present model. The difference originates from the elastic energy of the slab that is still supported by the weak layer. Because Heierli models only the unsupported section of the slab, which can be thought of as a rigid weak layer, this energy contribution is neglected (Figure 1).

Aside from the improved accuracy of the energy release rate of cracks, the contribution of the present model is significant because it provides weak layer stresses in the same analysis. One might again argue that while slab and weak layer are rather homogeneous, the elastic halfplane solution shown by Föhn's [17] suffices. However, i) weak layers that are softer than a bonded slab are characteristic for skier-triggered avalanche events [18,19,20] and ii) the presented modeling strategy allow for considering arbitrarily layered slabs instead of just homogeneous ones. That is, it is capable of providing analytical expression for both weak layer stress and the energy release rates of cracks in stratified snowpacks. This is important because, for instance, melt-freeze crusts can render slabs stiff in bending yet soft in tension depending on their location within the snowpack. A corresponding follow-up work is already in preparation.

[1] J. Heierli and M. Zaiser. Failure initiation in snow stratifications containing weak layers: Nucleation of whumpfs and slab avalanches. Cold Regions Science and Tech-

nology, 52(3):385–400, 2008.

[2] C. Scapozza. Entwicklung eines dichte- und temperaturabhängigen Stoffgesetzes zur Beschreibung des visko-elastischen Verhaltens von Schnee. PhD thesis, ETH Zürich, 2004.

[3] J. Heierli. Anticrack model for slab avalanche release. PhD thesis, Universität Karlsruhe, 2008.

[4] J. Schweizer, B. Reuter, A. van Herwijnen, B. Richter, and J. Gaume. Temporal evolution of crack propagation propensity in snow in relation to slab and weak layer properties. The Cryosphere, 10(6):2637–2653, 2016.

[5] J. Schweizer, B. Reuter, A. van Herwijnen, D. Gauthier, and B. Jamieson. On how the tensile strength of the slab affects crack propagation propensity. In Proceed- ings, International Snow Science Workshop, Banff, pages 164–168, 2014.

[6] B. Reuter, J. Schweizer, and A. Van Herwijnen. A process-based approach to estimate point snow instability. Cryosphere, 9(3):837–847, 2015.

[7] J. Gaume, A. van Herwijnen, G. Chambon, K. W. Birkeland, and J. Schweizer. Modeling of crack propagation in weak snowpack layers using the discrete ele- ment method. The Cryosphere, 9(5):1915–1932, 2015.

[8] J. Gaume, A. van Herwijnen, G. Chambon, N. Wever, and J. Schweizer. Snow fracture in relation to slab avalanche release: critical state for the onset of crack prop- agation. The Cryosphere, 11(1):217–228, 2017.

[9] L. Benedetti, J. Gaume, and J.-T. Fischer. A mechanically-based model of snow slab and weak layer fracture in the Propagation Saw Test. International Journal of Solids and Structures, 2018.

[10] B. Reuter, N. Calonne, and E. Adams. Shear failure of weak snow layers in the first hours after burial. The Cryosphere Discussions, (January):1–17, 2019.

[11] K. W. Birkeland, A. van Herwijnen, B. Reuter, and B. Bergfeld. Temporal changes in the mechanical properties of snow related to crack propagation after loading. Cold Regions Science and Technology, 159:142–152, 2019.

[12] B. Gerling, H. Löwe, and A. van Herwijnen. Measuring the Elastic Modulus of

Snow. Geophysical Research Letters, 44:11,088–11,096, 2017.

[13] B. Köchle and M. Schneebeli. Three-dimensional microstructure and numerical calculation of elastic properties of alpine snow with a focus on weak layers. Jour- nal of Glaciology, 60(222):705–713, 2014.

[14] P. M. B. Föhn. Simulation of surface-hoar layers for snow-cover models. Annals of Glaciology, 32:19–26, 2001.

[15] S. Horton, S. Bellaire, and B. Jamieson. Modelling the formation of surface hoar layers and tracking post-burial changes for avalanche forecasting. Cold Regions Science and Technology, 97:81–89, 2014.

[16] B. Jamieson and J. Schweizer. Texture and strength changes of buried surface-hoar layers with implications for dry snow-slab avalanche release. Journal of Glaciology, 46(152):151–160, 2000.

[17] P. M. B. Föhn. The stability index and various triggering mechanisms. In Avalanche Formation, Movement and Effects (Proceedings of the Davos Sym- posium, Sept. 1986), volume IAHS Publ., pages 195–214, 1987.

[18] J. Schweizer and B. Jamieson. Snowpack properties for snow profile analysis. Cold Regions Science and Technology, 37(3):233–241, 2003.

[19] J. Schweizer, B. Jamieson, and M. Schneebeli. Snow avalanche formation. Reviews of Geophysics, 41(4):1016, 2003.

[20] A. van Herwijnen and B. Jamieson. Snowpack properties associated with fracture initiation and propagation resulting in skier-triggered dry snow slab avalanches. Cold Regions Science and Technology, 50(1-3):13–22, 2007.

[Figure]

**Fig. 1.** Impact of model assumptions on slab deformations that directly affect the stored energy.

[Figure]

**Fig. 2.** Fig. 11 of the manuscript recomputed with slab Young's modulus E_slab = 5.23 MPa, weak layer thickness t = 1 cm and different weak layer Young's moduli E_weak.

---

## Author Response (AR2)

**Response to additional remarks by Michael Zaiser**

The reviewer agrees with our response to his initial concerns and asks for further clarification of the derivation of anticrack model equations we use as reference to benchmark our model. Below, we provide the requested details showing the reviewer's comments in blue. We thank the reviewer for his time and the detailed review.

The authors have addressed most criticisms raised in the previous report. However, I have one more major concern. Comparison with the work of Heierli (2008) is carried out in a unusual manner, since the equation (30) used by the authors in the present manuscript does not stem from Heierli but from a paper of Schweizer (2011). I have two concerns regarding this equation:

1) Schweizer (2011) give w0 in Eq. (30) as (3/4) eta^2 tau^2. I cannot easily see how the authors define w0. It seems to be explained in Figure 4 but I cannot see how this could match the expression of Schweizer. Please give a clear mathematical (not graphical) definitiion and clarify.

2) Eq. (4) of Schweizer (2011) , which equals Eq. (30) of the present paper, is supposed to follow from an original expression of Heierli (Eq. (1) of Schweizer 2011 or Eq. (4.13) in the thesis of Heierli). I cannot easily see how this derivation has been obtained (I tried to derive it but failed). Please give a derivation that shows how your Eq. (30) follows from the original relations given by Heierli.

I feel unable to asess the implications of the above two criticisms. If the requested derivations can be provided then I see no problem with publishing the paper as is. If they can not, then the anyalysis surrounding Eq. (30) and Figure 14 needs to be re-done from scratch.

In the work of Heierli (2008) the strain energy of the considered notched configuration is given. The differentiation of this energy with respect to the crack length provides the energy release rate of the anticrack model (Eq. (3) in Schweizer (2011)). This provides the closed-form analytical solution given in Schweizer (2011) in Eqs. (4)–(9). We have used this expression for the comparison in our work. However, as the reviewer pointed out, there are typos in the current version of our manuscript in the constants given in the appendix and $w_0$ is omitted. The corresponding paragraph now correctly reads:

With normal and shear loading $\bar{q}_\mathrm{n} = -\rho g h \cos\varphi$ and $\bar{q}_\mathrm{t} = \rho g h \sin\varphi$, respectively, the constants $w_0$ to $w_4$ of Eq. (30) read

$$w_0 = \frac{3\eta}{4}\bar{q}_\mathrm{t}^2, \tag{1}$$

$$w_1 = \left(\pi\gamma + \frac{3\eta}{2}\right)\bar{q}_\mathrm{t}^2 + 3\eta^2\bar{q}_\mathrm{n}\bar{q}_\mathrm{t} + \pi\gamma\bar{q}_\mathrm{n}^2, \tag{2}$$

$$w_2 = \bar{q}_\mathrm{t}^2 + \frac{9}{2}\eta\bar{q}_\mathrm{n}\bar{q}_\mathrm{t} + 3\eta\bar{q}_\mathrm{n}^2, \tag{3}$$

$$w_3 = 3\eta\bar{q}_\mathrm{n}^2, \tag{4}$$

$$w_4 = 3\bar{q}_\mathrm{n}^2, \tag{5}$$

in the notation of the present work where $\gamma \approx 1$ and $\eta = \sqrt{4\left(1+\nu\right)/5}$ are constants.

Also note that Fig. 12 misrepresents the boundary conditions used by Heierli. While it is correct that weak layer deformation is neglected in Heierli's model, there are no constraints imposed on the slab as shown in the figure.

We agree that Fig. 12 did not reflect the boundary conditions of the Heierli model correctly although closely matching Fig. 4.4 in the thesis of Heierli. The shear deformation $\psi$ at the boundaries of the anticrack interval is part of the energy functional of the Timoshenko beam and is not defined but obtained from energy minimization (discussed on p. 48 in the Thesis of Heierli). We have changed Fig. 12 to capture this correctly.